# A flow-based latent state generative model of neural population responses to natural images

**Mohammad Bashiri,[1,*] Edgar Y. Walker,[1,*] Konstantin-Klemens Lurz,[1]**
**Akshay Kumar Jagadish,[1,2] Taliah Muhammad,[3-4] Zhiwei Ding,[3-4] Zhuokun Ding,[3-4]**
**Andreas S. Tolias,[3-4] Fabian H. Sinz[5,1,†]**
[1] Institute for Bioinformatics and Medical Informatics, University of Tübingen, Germany
[2] Max Planck Institute for Biological Cybernetics, Tübingen, Germany
[3] Department for Neuroscience, Baylor College of Medicine, Houston, TX, USA
[4] Center for Neuroscience and Artificial Intelligence, Baylor College of Medicine, Houston, TX, USA
[5] Department of Computer Science, University Göttingen, Germany

[*]equal contribution, [†]`sinz@cs.uni-goettingen.de`

## Abstract

We present a joint deep neural system identification model for two major sources of neural variability: stimulus-driven and stimulus-conditioned fluctuations. To this end, we combine (1) state-of-the-art deep networks for stimulus-driven activity and (2) a flexible, normalizing flow-based generative model to capture the stimulus-conditioned variability including noise correlations. This allows us to train the model end-to-end without the need for sophisticated probabilistic approximations associated with many latent state models for stimulus-conditioned fluctuations. We train the model on the responses of thousands of neurons from multiple areas of the mouse visual cortex to natural images. We show that our model outperforms previous state-of-the-art models in predicting the distribution of neural population responses to novel stimuli, including shared stimulus-conditioned variability. Furthermore, it successfully learns known latent factors of the population responses that are related to behavioral variables such as pupil dilation, and other factors that vary systematically with brain area or retinotopic location. Overall, our model accurately accounts for two critical sources of neural variability while avoiding several complexities associated with many existing latent state models. It thus provides a useful tool for uncovering the interplay between different factors that contribute to variability in neural activity.

## 1 Introduction

Characterizing the activity of sensory neurons is a major goal of neural system identification. While neural responses in the visual cortex vary with visual stimuli, they also exhibit variability to the repeated presentations of identical stimuli [1–4]. This stimulus-conditioned variability has significant and sophisticated correlations among neurons commonly referred to as noise correlations [4–6] and exhibits dependency on various factors such as the stimulus [7–9], the behavioral task [10, 11], attention [12–14], and the general brain state [15, 16]. Understanding the nature of this correlated variability and its functional implication in the processing of sensory stimuli requires models that account for both stimulus-driven and shared stimulus-conditioned variability. The goal is thus to model the stimulus-conditioned response distribution $p(\mathbf{r}|\mathbf{x})$ of population activity $\mathbf{r} \in \mathbb{R}^n$ over $n$ neurons responding to an arbitrary sensory stimulus $\mathbf{x}$. However, models that account for stimulus-driven and stimulus-conditioned correlated variability have been developed largely independently.

35th Conference on Neural Information Processing Systems (NeurIPS 2021).

In the recent decade, we have seen significant progress in **modeling stimulus-driven activity**, largely driven by the use of deep neural networks (DNNs) [17–22]. Typically, the expected response of the neurons conditioned on the stimulus is captured as a function of the stimulus via a deep network $\mathbf{f}_\theta(\mathbf{x}) = \mathbb{E}[\mathbf{r}|\mathbf{x}]$ with learnable parameters $\theta$. These models can therefore predict how population responses depend on an arbitrary stimulus, and could even be used to derive stimuli that would yield desirable responses [23, 24]. Typically, these networks are trained using Poisson-loss, assuming that the population activity $\mathbf{r}$ is distributed around the stimulus-conditioned mean $\mathbf{f}_\theta(\mathbf{x})$ with an independent Poisson distribution. Therefore, existing state-of-the-art networks commonly ignore stimulus-conditioned correlations among neural responses, and impose strong assumptions about the form of the marginal distribution (i.e. Poisson) for each neuron. As sensory populations are known to exhibit noise correlations and deviate from Poisson distributions [4, 25, 26], this conditional independence assumption might limit the ability of these models to accurately capture $p(\mathbf{r}|\mathbf{x})$.

On the other hand, many of the existing **models for stimulus-conditioned variability** capture the variations in the population activity by specifically modeling the responses to repeated presentations of an identical stimulus. Many of these approaches employ statistical techniques such as maximum-entropy or copula distributions to reduce the number of parameters needed to fit the target distribution [27–29]. A popular approach has been to describe the stimulus-conditioned variability in terms of a typically lower-dimensional shared latent state $\mathbf{z}$: $p(\mathbf{r}|\mathbf{x}) = \int p(\mathbf{r}|\mathbf{x}, \mathbf{z})p(\mathbf{z}|\mathbf{x})\,d\mathbf{z}$ [16, 25, 26, 30–35]. Among these are hierarchical generative models that can capture more sophisticated relationships between the stimulus and noise correlations, as well as deviations from Poisson, such as over-dispersion [25, 26, 32, 34, 35]. While these approaches present powerful methods to capture stimulus-conditioned variability, they often fit $p(\mathbf{r}|\mathbf{x})$ separately for each unique stimulus and require responses to repeated presentations of the stimulus [16, 25, 26, 29, 35]. This limits their ability to yield predictions to a novel stimulus without requiring some stimulus-specific parameters to be learned. Furthermore, the increased complexity of the distribution usually requires a substantially more involved probabilistic machinery to make latent state inference and parameter fitting feasible. Consequently, most latent state models for neural data either ignore stimulus-driven variability altogether [30, 31, 34], or employ a very simple model of stimulus-driven variations [16, 25, 26, 32].

Here, we propose a new model that closes the gap between these two approaches by combining DNN-based models of stimulus-driven activity with a latent state model that accounts for shared stimulus-conditioned variability. While DNNs can be trained effectively via gradient-based optimization, the challenge is to avoid the complex probabilistic machinery associated with existing latent state models, particularly those that require stimulus-specific parameters to be learned over repeated presentations of identical stimuli. To this end, we combine normalizing flows [36–41] with Gaussian Factor Analysis (FA) models [42], where the stimulus-dependence occurs through a DNN that learns to shift the mean of the FA distribution based on the stimulus. FA models make use of multivariate Gaussian distributions with a particular low-rank structure of the covariance matrix. While the use of FA in capturing shared variability greatly simplifies inference and learning, it is not directly applicable to neural responses because neural responses are not Gaussian-distributed, particularly for low firing rates. To circumvent this problem, variance-stabilizing transformations, such as the square-root function, have been used in the past to make the responses more Gaussian-distributed [16, 30]. However, there may be other transformations that capture the response distribution more accurately. Furthermore, since the transformation for one neuron may not be applicable to other neurons, ideally it would be learned for each neuron separately. To achieve this flexibility, we allow our model to learn neuron-specific transformations with a marginal normalizing flow.

Normalizing flow models are density estimators that use a series of diffeomorphisms to transform the source density underlying the data into a simple distribution—typically an isotropic Gaussian of the same dimension. These transformations are usually chosen to have efficient-to-compute log-determinants, and typically act on the entire variable vector to capture any statistical dependencies between the dimensions. Here, we replace the isotropic Gaussian with an FA model to capture dependencies among dimensions and only use diffeomorphisms that act on each dimension separately, i.e. apply flow-based transformations on the marginals only. While this choice places certain restrictions on the complex dependencies between neurons that may be captured (refer to section 4 Discussion for details), it has two important advantages: (1) The generative model is easy to train while combining state-of-the-art deep networks with flexible latent state models, and (2) the use of marginal flows allows for an easy mechanism to compute conditional distributions of one neuron given responses of other neurons that would not be easy to obtain with non-marginal flow models.

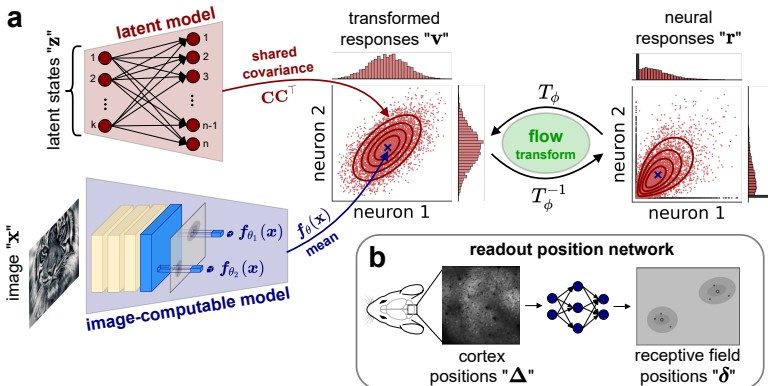

Figure 1: Flow-based Factor Analysis model. **a:** Schematic of the flow-based model relating all relevant variables in the study. **b:** Schematic of the sub-network used by the image-computable model to map cortical positions into receptive field positions. Refer to section 2 Methods for the details.

In summary, we make the following contributions. We (1) combine state-of-the-art DNN-based models with flow-based latent state models to jointly account for stimulus-driven and shared stimulus-conditioned variability in neural population activity. Our model can predict the distribution of neural responses to unseen stimuli, without the need for repeated presentations to learn stimulus-conditioned variability. We (2) apply our method on the activity of thousands of neurons in response to natural images, recorded via two-photon Calcium imaging from multiple areas of the mouse visual cortex. We demonstrate that our model outperforms current state-of-the-art methods in capturing the distribution of responses. Finally, we (3) show that our model infers latent state structures with meaningful relations to behavioral variables such as pupil dilation as well as other functional and anatomical properties of visual sensory neurons.

## 2 Methods

### 2.1 Models

**Flow-based Factor Analysis model (FlowFA)** For a given stimulus $\mathbf{x}$ and population response $\mathbf{r} \in \mathbb{R}^n$, where $n$ is the number of neurons, we define our normalizing flow-based Factor Analysis (FlowFA) model of the stimulus-conditioned population activity $p(\mathbf{r}|\mathbf{x})$ as

$$p(\mathbf{r}|\mathbf{x}, \theta, \phi) = \mathcal{N}(T_\phi(\mathbf{r}); \mathbf{f}_\theta(\mathbf{x}), \mathbf{C}\mathbf{C}^\top + \Psi) \cdot |\det\nabla_\mathbf{r} T_\phi(\mathbf{r})| \,. \tag{1}$$

FlowFA has two major parts: (1) A flow model $T_\phi$ with learnable parameters $\phi$ that transforms the population responses $\mathbf{r}$ such that the transformed responses $\mathbf{v} = T_\phi(\mathbf{r})$ are well modelled by a (2) Gaussian Factor Analysis (FA) model $\mathcal{N}(\mathbf{v}; \mathbf{f}_\theta(\mathbf{x}), \mathbf{C}\mathbf{C}^\top + \Psi)$ (Fig. 1a). Here, $\mathcal{N}(\mathbf{v}; \mu, \Sigma)$ denotes a Gaussian distribution over $\mathbf{v}$ with mean $\mu$ and covariance $\Sigma$. According to the FA model, the random variable $\mathbf{v}$ is generated via $\mathbf{v} = \mathbf{f}_\theta(\mathbf{x}) + \mathbf{C}\mathbf{z} + \varepsilon$ where $\mathbf{z} \in \mathbb{R}^k$ is a low-dimensional latent state with $k \ll n$ and an isotropic Gaussian prior $\mathbf{z} \sim \mathcal{N}(0, I_k)$ whose samples map to $\mathbf{v}$ via the *factor loading matrix* $\mathbf{C} \in \mathbb{R}^{n \times k}$. The effect of the stimulus $\mathbf{x}$ on the responses is captured by the mean of the FA distribution that depends on the stimulus, modeled as a deep network $\mathbf{f}_\theta(\mathbf{x}) \in \mathbb{R}^n$ with learnable parameters $\theta$ (Fig. 1a,b). We further include neuron-specific, independent noise $\varepsilon \sim \mathcal{N}(0, \Psi)$ where $\Psi \in \mathbb{R}^{n \times n}$ is a diagonal covariance matrix.

Since the flow model is a trainable change of variables, it introduces the absolute determinant $|\det\nabla_\mathbf{r} T_\phi(\mathbf{r})|$ of the Jacobian $\nabla$ of $T_\phi$ with respect to $\mathbf{r}$ into Eq. (1). The transform itself is a diffeomorphism, i.e. an invertible differentiable mapping $T_\phi : \mathbb{R}^n \mapsto \mathbb{R}^n$ allowing us to evaluate the exact likelihood of each data point and easily draw samples from the model. Therefore, the model serves as a fully generative model from which samples of the stimulus-conditioned population responses can easily be generated for an arbitrary stimulus.

In the model formulation presented here, we choose $T_\phi$ to act on each single dimension separately, i.e. $T_\phi(\mathbf{r}) = [T_{\phi_1}(r_1), ..., T_{\phi_n}(r_n)]^\top$. This choice results in a diagonal Jacobian which not only substantially simplifies the form of the determinant to $\det\nabla_\mathbf{r} T_\phi(\mathbf{r}) = \prod_{i=1}^n \frac{\partial T_{\phi_i}}{\partial r_i}$, but also allows us to easily compute conditionals and marginals (see appendix A for the details). This would not generally be possible for diffeomorphisms with a non-diagonal Jacobian.

**Zero-Inflated Flow-based Factor Analysis model (ZIFFA)** For two-photon Calcium imaging, a significant portion of inferred neural activity is zero, resulting in a sharp peak at zero in the response distribution (i.e. zero-inflated distribution) [43]. This zero-inflation is potentially a problem for the FlowFA model since the model would attempt to generate the peak at zero by mapping a large proportion of the Gaussian probability mass onto the "zero" responses, resulting in a poor fit to the response distribution. To avoid this, we extend FlowFA by modeling the zero responses with a separate peak (similar to Wei et al. [43]) and applying the FlowFA model to capture only the positive responses. We refer to this model as Zero-Inflated Flow-based Factor Analysis (ZIFFA). More specifically, ZIFFA is a mixture model that models neural responses below and above a threshold value $\rho$ with two separate, non-overlapping distributions. To capture the peak at zero, the responses below the threshold (i.e. "zero" responses) are modeled by a uniform distribution, while FlowFA is used to capture responses above the threshold:

$$p(\mathbf{r}|\mathbf{x}) = \left( \prod_{\{i:r_i \leq \rho\}} \frac{1 - q_i(\mathbf{x})}{\rho} \right) \cdot \left( \prod_{\{i:r_i > \rho\}} q_i(\mathbf{x}) \right) \cdot \mathcal{N}(T_\phi(\mathbf{r}_+); f_{\theta,+}(\mathbf{x}), \mathbf{C}_+\mathbf{C}_+^\top + \Psi_+) \cdot |\nabla T_\phi(\mathbf{r}_+)|,$$

(2)

where $q_i(\mathbf{x})$ is the probability of the response being above the threshold $\rho$ modeled, jointly with the mean of the FA, as a function of the stimulus via a DNN $\mathbf{f}_\theta$ with learnable parameters $\theta$. $\mathbf{r}_+$ and $f_{\theta,+}(\mathbf{x})$ are the sub-vectors, and $\mathbf{C}_+$ and $\Psi_+$ are the sub-matrices corresponding to responses above the threshold, and $\theta, \mathbf{C}, \Psi$ are the same as defined in Eq. (1). Refer to appendix B for the derivation.

**Control models** We compare the FA-based models against two control models used for neural system identification that assume independence among neurons with specific forms of marginal distributions inspired by existing work: (1) Poisson [18, 22] and (2) Zero-inflated Gamma (ZIG) [43]. To capture continuous neural responses measured with Calcium imaging, we relax the discrete Poisson distribution into a continuous distribution by assuming $r = \hat{r} + \epsilon$ where $\hat{r} \sim \text{Poisson}(\lambda)$ and $\epsilon \sim \text{Uniform}[0, 1)$. This yields the likelihood function

$$p_{\text{poiss}}(\mathbf{r}|\mathbf{x}) = \prod_i^n \frac{\lambda_i(\mathbf{x})^{\lfloor r_i \rfloor} e^{-\lambda_i(\mathbf{x})}}{\lfloor r_i \rfloor!},$$

(3)

where $\lambda(\mathbf{x}) = \mathbf{f}_\theta(\mathbf{x})$ is the predicted firing rate of the neurons to input image $\mathbf{x}$ modeled as a DNN $\mathbf{f}_\theta$ with learnable parameters $\theta$. The ZIG distribution is a mixture of a uniform and a gamma distribution separated at the value $\rho$ with no overlap [43]:

$$p_{\text{ZIG}}(\mathbf{r}|\mathbf{x}) = \prod_i^n \left( \frac{1 - q_i(\mathbf{x})}{\rho} + \frac{q_i(\mathbf{x}) r_i^{\kappa_i - 1}}{\Gamma(\kappa_i) \nu_i(\mathbf{x})^{\kappa_i}} \exp\left( -\frac{r_i}{\nu_i(\mathbf{x})} \right) \right),$$

(4)

where $\nu_i(\mathbf{x})$ is the scale parameter of the gamma distribution, and $q_i(\mathbf{x})$ is same as in Eq. (2). To formulate ZIG as an image-computable model, $\nu_i(\mathbf{x})$ and $q_i(\mathbf{x})$ are jointly modeled using a DNN $\mathbf{f}_\theta$ with learnable parameters $\theta$. Similar to Wei et al. [43], we let the shape parameter $\kappa_i$ be neuron-specific, but independent of the input. Importantly, we used the same value for $\rho$ in both ZIG and ZIFFA models.

Note that when the covariance matrix of the FA-based models is diagonal (i.e. 0-dimensional latent state), these models assume independence among neurons and their performance is directly comparable to the control models.

## 2.2 Model components

**Deep convolutional neural network $\mathbf{f}_\theta$** We capture the stimulus-driven changes in the neuronal response distribution using a deep convolutional neural network $\mathbf{f}_\theta(\mathbf{x})$ with the same architecture as used by Lurz et al. [22]. Briefly, the network consists of two parts: (1) A shared four-layer core network, where each layer consists of a standard or depth-separable [44] convolution operation resulting in 64 feature channels, followed by batch normalization and ELU nonlinearity, and (2) a neuron-specific readout mechanism (referred to as "Gaussian readout") that learns the position of the neuron's receptive field (RF) and computes a weighted sum of the features at this position along the channel dimension (Fig. 1a). In contrast to Lurz et al. [22] where the RF positions $\boldsymbol{\delta}$ in image space were obtained by applying a shared affine transformation on the experimentally measured cortical positions $\boldsymbol{\Delta}$ of the neurons, here we allow this mapping to take on a non-linear form to allow flips

in the representation of the visual field as a function of cortical position (Fig. 1b). This is crucial to model cortex-to-visual space mappings for multiple brain areas, as the retinotopy of some areas are mirrored with respect to each other. During training, we apply L1 regularization to the readout feature weights and L2 regularization on the Laplace-filtered weights of the first convolution layer.

**Normalizing flow** $T_\phi$   We construct the marginal flow model $T_\phi =$ affine $\circ$ exp $\circ$ affine $\circ$ ELU $\circ$ affine $\circ$ ELU $\circ$ affine $\circ$ log $\circ$ affine from a set of monotonic functions $\{$ affine, ELU, log, exp$\}$, of which only the affine transformation has learnable parameters. We restricted all the affine transformation layers to have positive scale, and additionally restricted the first affine layer to have a positive offset. For each neuron indexed by $i$, we learn a separate marginal transformation $T_{\phi_i}$. We compare the flow transformation against two common fixed transformations: square-root [16, 30] and Anscombe [45]. These two transformations can be expressed by the general form $u = \exp(a \log(y + b) + c)$ which is a series of affine, log, affine, and exp transformations, with $a = 0.5$, $b = 0$, and $c = 0$ for square-root, and $a = 0.5$, $b = \frac{3}{8}$, and $c = \log(2)$ for Anscombe. We specifically chose the components of $T_\phi$ such that these common fixed transformations exist as special cases, ensuring that the flow transformations are strictly more flexible than any choice of fixed transformations commonly found in the literature. For ZIFFA, we adjusted the formulation of the marginal flow $T_\phi$ such that the predicted neuronal responses remain above $\rho$, the boundary between the uniform and the FlowFA components of the mixture model, by replacing the first affine transformation in $T_\phi$ with a layer that only shifts by $-\rho$.

### 2.3   Neural and behavioral data

We recorded the response of neurons in mouse visual cortices (layer L2/3) to gray-scale natural images using a wide-field two-photon microscope [46] (see appendix C for details). In this study, we used two scans from two mice spanning three visual areas: primary visual cortex (V1) and lateromedial area (LM) in scan 1; V1 and posteromedial area (PM) in scan 2. A total of 2,867 V1 neurons and 907 LM neurons were recorded in scan 1; 5,029 V1 neurons and 3,343 PM neurons were recorded in scan 2. Among these, we used 1,000 V1 and 907 LM neurons from scan 1, and 1,000 V1 and 1,000 PM neurons from scan 2. For both scans, neurons were randomly selected if the area contained more than 1,000 neurons. We also recorded behavioral variables such as pupil dilation, simultaneously. The natural image stimuli were sampled from ImageNet [47], cropped to fit a monitor with 16:9 aspect ratio, and presented to the mice at a resolution of $0.53$ ppd (pixels per degree of visual angle). A total of 6,000 images were shown in each scan, of which 1,000 images consist of 100 unique images each repeated 10 times to allow for an estimate of the neural response variability. We used the repeated images for testing, and split the remaining images into 4,500 training and 500 validation images.

### 2.4   Model fitting and evaluation

**Fitting**   We trained all models end-to-end via gradient-based optimization to maximize the log-likelihood obtained from Eqs. (1), (2), (3) or (4) for the corresponding model, optimizing over all learnable parameters. To ensure that $\Psi$, the diagonal covariance matrix, stays positive-valued, we re-parameterized $\Psi = e^\nu$ and optimized $\nu$ instead. To find the best image-computable DNN models, we used Bayesian optimization [48] to find hyper-parameters that maximized the final log-likelihood of the trained model. Hyper-parameters include the learning rate and regularization coefficient on the readout weights. The log-likelihood used for scheduling learning rate, early stopping, and finding hyper-parameters was computed on the validation set. Additional details about training can be found in appendix D. The code can be found at `https://github.com/sinzlab/bashiri-et-al-2021`.

**Evaluation**   We compared the FA-based models (ZIFFA, FlowFA, and FA with fixed transformations) to the control models based on likelihood and leave-neuron-out prediction correlation on the test set. For the former, we computed the likelihood of the responses in bits per neuron per image under each model, based on Eqs. (1), (2), (3), and (4), accordingly. For the correlation measure, we computed the Pearson correlation between the predicted and the measured responses of each neuron on the test set. For the FA-based models that may capture the statistical dependency (i.e. covariance) between neurons, we predicted the response of a given neuron conditioned on the responses of all other neurons recorded simultaneously on the trial. More specifically, given an image $\mathbf{x}$ and the response of all other neurons $\mathbf{r}_{\setminus i}$, we estimated the response of a neuron $r_i$ to the image by computing the posterior mean of the neuron's response $\mathbb{E}[r_i | \mathbf{x}, \mathbf{r}_{\setminus i}]$. We refer to this measure as *conditional correlation* (see appendix E for details).

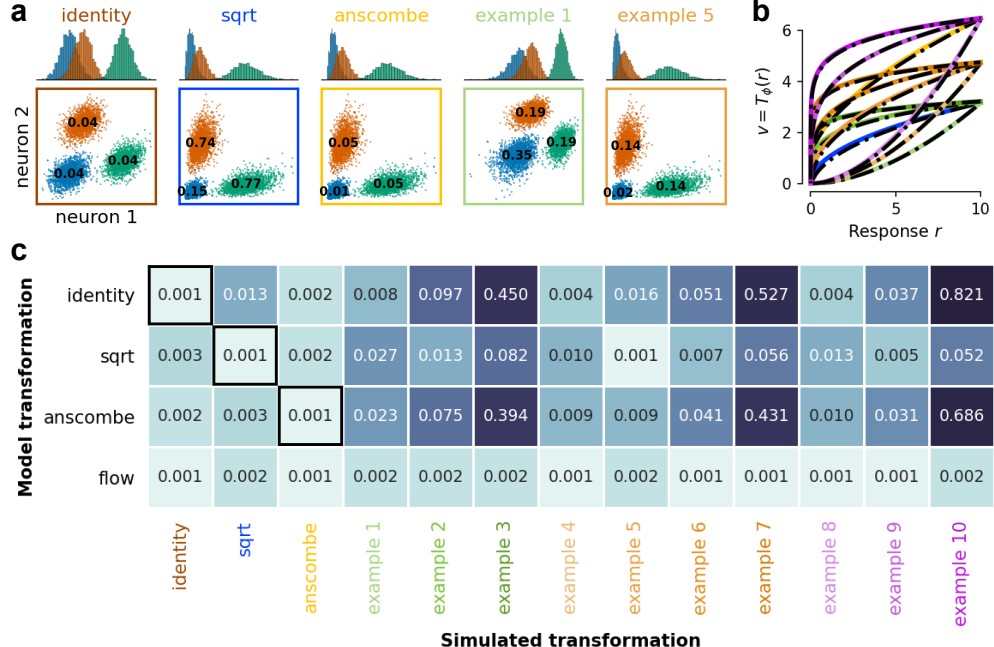

Figure 2: FlowFA model recovers the underlying transformation. **a:** Simulated responses for 2 neurons under various transformations. Across all transformations, *transformed* responses were sampled from Gaussian distributions with differing means (indicated by the color of the samples) but identical covariance. The covariance between the two neurons is shown in black text. **b:** Transformations learned by the flow model are shown in black, overlaid on the ground-truth transformations. **c:** Performance of models with fixed or learned (flow) transformations (rows) trained on responses simulated with a variety of transformations (columns). Cases where the simulating and trained transformations are the same are indicated by black outlines. Performance is measured as the KL divergence between the modeled and ground-truth distributions, where 0 would correspond to a perfect fit.

## 3 Results

### 3.1 Model performance

**FlowFA model faithfully recovers invertible transformations on synthetic data** We first used synthetic data to illustrate that our FlowFA model with a learnable transformation can adequately learn and recover a wide variety of transformations resulting in different response distributions. To this end, we sampled 5,000 data points for 100 neurons from models with different ground-truth transformations (see appendix F for details on data generation). The invertible transformations (Example 1–10) had the general form $\exp(a \log(y - b) + c)$ with differing values of $a$, $b$, and $c$ (Fig. 2b). We trained FA-based models with either a fixed (FixedFA) or a learnable flow-based (FlowFA) transformation. As expected, the models with a fixed transformation performed well if the data was generated with a similar transformation, but the performance suffered when the transformations differed (Fig. 2c, first three rows). In contrast, the FlowFA model was able to flexibly learn every underlying transformation (Fig. 2b) and effectively captured all distributions across all simulations (Fig. 2c, last row).

**Flow-based models capture cortical response distribution well** After demonstrating that the flow-based model can effectively fit a wide range of distributions, we used it to capture distributions of the mouse visual cortex population responses to natural images, recorded in two different two-photon scans from two mice (scan 1 and scan 2, refer to section 2.3 for details). We trained the FA-based models (ZIFFA, FlowFA, and FixedFA) for different values of latent dimensions $k \in \{0, 1, 2, 3, 10\}$. We measured the model performance by computing the log-likelihood as well as the conditional correlations (see section 2.4).

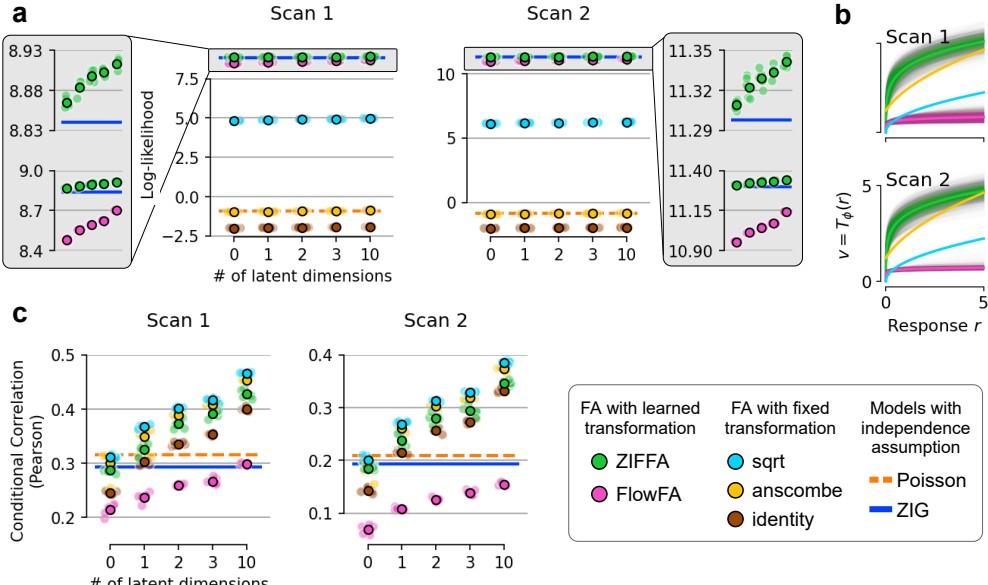

Figure 3: Comparison of models trained on the mouse visual cortical population responses to natural images. **a**: log-likelihood computed for models trained on scan 1 (left panel) and scan 2 (right panel). Values for both individual (lighter shade) and average (darker shade) performance of a model trained under various random seeds are shown. Gray block provides a zoomed-in view of the ZIFFA, FlowFA, and Zero-Inflated-Gamma (ZIG) models. **b**: Neuron-specific transformations learned by the flow-based models (ZIFFA in green, average across neurons in light green; FlowFA in pink, average across neurons in light pink) shown in comparison to fixed transformations. **c**: Conditional correlation. Format is similar to **a**.

The ZIFFA model outperformed all other models across all numbers of latent dimensions $k$ in terms of log-likelihood (Fig. 3a). Furthermore, with increasing latent dimensions, the conditional correlation of the ZIFFA model improved significantly beyond the control models (Fig. 3c). Interestingly, we observed that the ZIFFA model exhibited slightly lower correlation performance compared to models with fixed transformations, reflecting that fitting models on likelihood does not necessarily yield optimal correlation. Importantly, the flow-based models outperformed all FixedFA models in terms of likelihood, which is corroborated by the fact that the learned transformation markedly differs from all fixed transformations and from one neuron to the other (Fig. 3b). Overall, the results suggest that the ZIFFA model is able to capture the (marginal) neural response distributions more accurately than other models (Fig. S2) while at the same time it learns and takes advantage of the statistical dependencies between neurons.

## 3.2 Uncovering biological insights from the trained model

Here, we explore the utility of our model in uncovering potential biological insights. All analyses were performed on the trained ZIFFA model with 3 latent dimensions.

**Model-based visual area identification** Several visual areas in mice show retinotopies that are "flipped" with respect to each other [49]. Intuitively, this means that if a point moves along the cortical surface, as it crosses the boundary between two "mirrored" areas, its counterpart in visual space would reverse its movement direction. As described in section 2.2, our model is equipped with a component network that predicts the RF location $\delta$ of each neuron in visual space as a function of its cortical location $\Delta$ (Fig. 1b). This network can be used to infer distinct visual cortical areas by detecting where the retinotopy "flips" with respect to the cortical position. To detect this flip we looked at the sign of the determinant of the Jacobian of the RF positions with respect to cortical positions $\det \frac{\partial \delta}{\partial \Delta}$. The sign can detect changes in the direction because (1) the sign of a determinant flips if one of the column or row vectors of the Jacobian matrix flips and (2) the determinant is invariant under rotation. When we compare distinct areas identified via the model to the experimentally identified areas, we find a very good match (Fig. 4a, left vs. right panels). To assess the quality of the learned

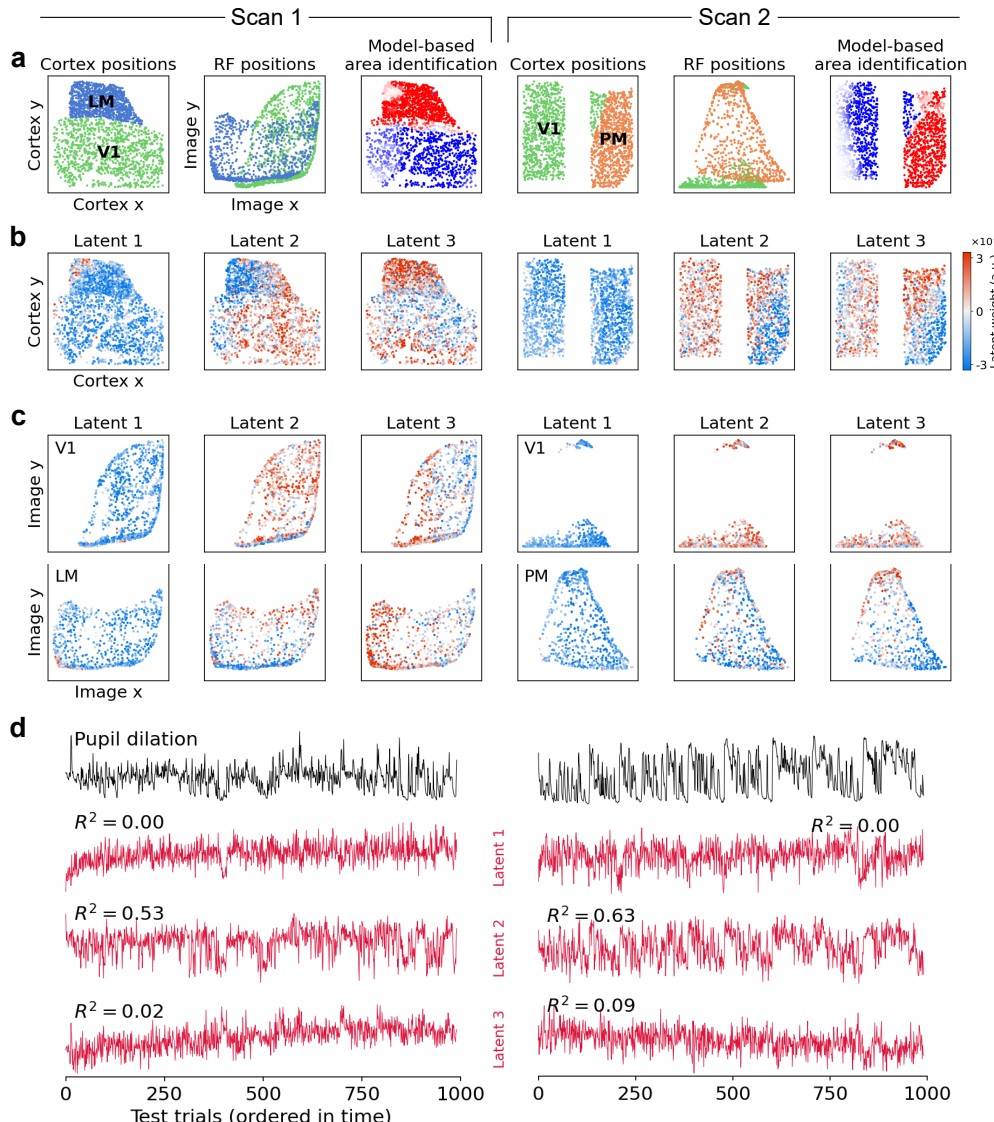

Figure 4: Analysis of the ZIFFA model with 3-dimensional latent state ($k = 3$). **a**: Model-based area identification from responses of visual sensory neurons to natural images. Left panel (Cortex positions): cortical position of the recorded neurons color-coded by experimentally identified areas (green: V1; blue: LM; orange: PM). Middle panel (RF positions): learned receptive field position for each neuron as a function of cortical positions color-coded by experimentally identified areas. Right panel (Model-based area identification): visual areas identified via the model by computing the determinant of the relative changes in RF position with respect to changes in cortical position; blue color shows negative determinant (i.e. mirrored visual field representation) and red color shows positive determinant (i.e. non-mirrored visual field representation). **b–c**: Distribution of the latent-to-neuron weights across cortical positions (**b**) and receptive field positions (**c**). **d**: Pupil dilation (black) and the inferred latent states (red) across trials from the test set. $R^2$ values are computed between the inferred latent state and the pupil dilation.

mapping, we quantified how well our model can identify distinct visual brain areas via the sign of the determinant. Across models initialized and trained with different random seeds, the sign correctly classifies distinct brain areas with an accuracy of $84\% \pm 3.4\%$ (SEM) and $75\% \pm 7.7\%$ (SEM). Because the experimental methods to determine area assignment that we use as ground truth can be quite coarse, the actual accuracy could even be higher. This suggests that our model could in principle

allow neuroscientists to identify distinct visual areas from responses to natural images alone, without the need for an extra experiment for area identification.

**Inferred latent states and their functional and anatomical implications** We next explored the latent states and how they relate to anatomy or behavior. For any particular trial, the FA-based models allow us to infer the most probable latent state $\mathbf{z}$ (MAP estimate), where the effect of each latent dimension on the neural population is captured by the factor loading matrix $\mathbf{C}$. However, as formulated in Eq. (1) and (2), interpreting the inferred latent states $\mathbf{z}$ can be difficult because the latent dimensions can be arbitrarily permuted and rotated (with corresponding changes in $\mathbf{C}$) without affecting the fit of the model. To facilitate interpretability of the inferred latent states, we follow a similar procedure used by Yu et al. [30] to extract *orthonormalized latent states* which are uniquely ordered by the amount of response variability each latent dimension accounts for (see appendix G for detailed explanation).

The orthonormalized latent states inferred from the ZIFFA model showed strong correlations with behavioral variables such as pupil dilation (Fig. 4d), as expected from previous works that use pupil dilation as a proxy for arousal and attention [50–54]. Interestingly, pupil dilation correlated most strongly with the second latent dimension in both scans with $R^2$ values of 0.53 ($p < 0.001$, two-tailed test for significance of correlation [55]) and 0.63 ($p < 0.001$) for scan 1 and scan 2, respectively, comparable to values previously reported [56]. To our surprise, this observation was consistent across models initialized and trained with different random seeds (Fig. S4b). To further quantify how well the latent states can jointly predict the pupil dilation, we regressed the pupil dilation against the latent states (Fig. S4a). The resulting $R^2$ values were 0.56 ($p < 0.001$) and 0.76 ($p < 0.001$) for scan 1 and scan 2, respectively. The high correlation between the latent states and the known surrogates of global brain state such as pupil dilation suggests that the latent model is able to learn meaningful dependencies and common factors in neural population.

Next, we explored whether the effect of the orthonormalized latent states on the neurons is related to their cortical or RF positions. To this end, we plotted the sign and magnitude of the weight mapping from the latent state to each neuron on the cortical position (Fig. 4b) or the RF positions of the neurons (Fig. 4c). We observed that the effect of some latent dimensions vary systematically across brain areas where the latent dimension has generally opposite effect on different areas (Fig. 4b: dimension 2 for both scans). In addition, some latent dimensions seemed to vary as a function of RF positions/retinotopy where a differential effect of the latent dimension is observed for both areas (Fig. 4c: dimension 3 for both scans). Interestingly, the first dimension which accounts for most of the shared variability in neural responses (refer to section G for more details) seemed to have a global effect that does not vary across different visual areas. These observations illustrate that our model can be a useful tool for uncovering the functional and structural implications of the behavioral or internal processes associated with the inferred latent states.

While the result of the analyses we present here are promising, we would like to point out that all analyses are preliminary, and conclusive biological interpretations would require additional rigorous experiments and analyses.

## 4 Discussion

**Getting the best of both worlds** Two major components of the variability in the activity of cortical neurons are the variability due to stimulus and the variability due to unobserved or internal processes, such as behavioral tasks or general brain states, that affect population of neurons in similar ways giving rise to correlated variability among neurons. Here, we presented a model that combines state-of-the-art DNN-based models to predict stimulus-driven changes in neural activity with a simple, yet flexible, flow-based factor analysis model to account for correlated neural activity. This formulation allows us to evaluate the exact likelihood of neural responses, easily sample stimulus-conditioned responses, and efficiently compute conditional and marginal distributions of subsets of neurons. By fitting this model to the activity of thousands of neurons from multiple areas of mouse visual cortex in response to natural images, we obtained state-of-the-art performance in capturing neural response distribution while additionally yielding latent states that exhibit meaningful relations to anatomy and functional properties of visual sensory neurons.

**Modeling zero-inflated response distribution** Flow models use diffeomorphisms to map one distribution into another. However, diffeomorphisms cannot transform a single peak at 0—typically

observed in neural responses recorded via Calcium imaging—into a smooth distribution such as Gaussian used in our model. The ZIFFA model avoids this problem by only transforming the positive part of the response with a diffeomorphism while explicitly capturing the peak at 0 via a uniform distribution as found in ZIG. Importantly, ZIFFA preserves all properties of the FlowFA model, while capturing the marginal distributions more accurately (Fig. S2), achieving a higher likelihood (Fig. 3), and learning more consistent and less step-like transformations (Fig. S3).

**Dependency of noise correlation on the stimulus**  The presented flow-based models learn a nonlinear transformation between a simple distribution (Gaussian FA) and the neural response distribution. While the learned covariance structure on the "transformed" neural responses captured by the FA model does not vary with the stimulus and the stimulus is only used to shift the mean of the FA model, this is not true for samples from the FA model transformed back into "neural response space" because the nonlinear flow transformation can introduce changes in the covariance as the mean varies (Fig. 2a). This mean-dependent change in the covariance potentially allows the model to capture changes in the covariance structure based on stimulus through the nonlinear transformation. A possible extension of our model is an explicit dependence of the FA's covariance matrix on the stimulus, which would allow the model to capture more complex dependencies between the stimulus and covariance structure.

**Comparison to related methods**  Our approach in capturing stimulus-conditioned variability is related to many existing approaches, or can be seen as a generalization thereof, while being computationally easier to handle at the same time. Recently, Keeley et al. [35] captured the trial-by-trial fluctuations by modeling the stimulus-specific and trial-specific latents via Factor Analysis (FA) models much like in our model. Importantly, while we capture the dependence of the stimulus-specific latents on the stimulus explicitly via a trained DNN, they inferred it from repeated presentations of the stimulus. Furthermore, the final Poisson distribution used to map from the latents to the distribution of neurons can be captured in our model via the flow-based transformation (e.g. inverse Anscombe) that maps Gaussian-distributed latents into a continuous approximation of a Poisson distribution. Moreover, the use of FA in combination with the marginal flow makes our approach related to copula-based distribution approximation and related approaches [28, 29, 57]. However, by explicitly limiting the stimulus dependence to occur via the shift in the mean of the FA model along with flow-based transformation of responses, we avoid the reliance on the repeated presentations of the stimuli [29] or highly constrained forms of the marginal distribution [28].

**Limitations and future extensions**  As discussed above, our flow-based approach generalizes several existing methods to capture stimulus-conditioned variability of neural responses while being computationally more tractable. This allows us to train our models end-to-end directly on the likelihood via common gradient-based optimization algorithms. Within this general framework, we presented a specific case where we learned neuron-specific stimulus-independent transformations, mapping responses into a FA model whose mean varies with the stimulus. As noted earlier, for each stimulus, this approach closely parallels Gaussian copula and thus shares much of the same limitations. Also, the fact that stimulus-dependent changes in the covariance structure only occur through the learned transformation implies that the model can only capture changes in the covariance structure that varies with the mean (a limitation shared with many of the existing models). That being said, we believe that our general approach of flow-based modeling of neural response distributions allows for several generalizations that would overcome these limitations. Examples include an explicit dependence of the FA's covariance matrix on the stimulus, as well as the usage of richer, potentially stimulus-dependent, learnable transformations.

**Broader impact**  Accurate models of neural variability such as the one presented here can lead to deeper scientific insights and understanding of how brains perceive and compute with sensory information, and can eventually also provide insights into how neurological and psychological disorders may disturb these functions. In particular, a more accurate model that relates internal brain states, stimulus-driven responses, and anatomical features such as retinotopy or memberships to certain brain areas might provide deeper insights into the computational principles of cortex. Naturally, our model requires data from animal experiments to be trained. However, we used existing datasets with very general protocols that can be used in several analyses to make efficient scientific use of data from animal experiments. Furthermore, models such as the one presented here do help to reduce the amount of animal experiments as faithful models allow us to explore the functional principles of neural populations *in silico*.

## Acknowledgments and Disclosure of Funding

We thank all reviewers for their constructive and thoughtful feedback. Furthermore, we thank Alexander Ecker, Konstantin Willeke, Arne Nix, Christoph-Benjamin Blessing, and Emmanouil Froudarakis for comments and discussions. MB is supported by the International Max Planck Research School for Intelligent Systems. KKL is funded by the German Federal Ministry of Education and Research through the Tübingen AI Center (FKZ: 01IS18039A). FHS is supported by the Carl-Zeiss-Stiftung and acknowledges the support of the DFG Cluster of Excellence "Machine Learning – New Perspectives for Science", EXC 2064/1, project number 390727645. This work was supported by an AWS Machine Learning research award to FHS. Also supported by the Intelligence Advanced Research Projects Activity (IARPA) via Department of Interior/Interior Business Center (DoI/IBC) contract number D16PC00003. The U.S. Government is authorized to reproduce and distribute reprints for Governmental purposes notwithstanding any copyright annotation thereon. Disclaimer: The views and conclusions contained herein are those of the authors and should not be interpreted as necessarily representing the official policies or endorsements, either expressed or implied, of IARPA, DoI/IBC, or the U.S. Government.

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
