## A  Expression for the marginal and conditional distributions

Here we derive and show that the marginal and conditional distributions in the neural response space can be straightforwardly expressed in terms of the corresponding marginal and conditional distributions in the transformed response space when the transformation function $T$ is separable. Consider partitioning neurons into two mutually-exclusive subgroups $\mathbf{r}^{(1)}$ and $\mathbf{r}^{(2)}$. Furthermore assume that the transformation function factorizes over these two subgroups such that $T(\mathbf{r}) = [T_1(\mathbf{r}^{(1)})^\top, T_2(\mathbf{r}^{(2)})^\top]^\top = [\mathbf{v}^{(1)\top}, \mathbf{v}^{(2)\top}]^\top = \mathbf{v}$, for some constituent diffeomorphisms $T_1$ and $T_2$. Given this,

$$
p_r\left(\mathbf{r}|x\right) = p_r\left(\mathbf{r}^{(1)}, \mathbf{r}^{(2)}\Big|x\right)
$$
$$
= p_v\left(T_1\left(\mathbf{r}^{(1)}\right), T_2\left(\mathbf{r}^{(2)}\right)\Big|x\right) \cdot \left|\det\nabla_{\mathbf{r}^{(1)}} T_1\left(\mathbf{r}^{(1)}\right)\right| \cdot \left|\det\nabla_{\mathbf{r}^{(2)}} T_2\left(\mathbf{r}^{(2)}\right)\right|,
$$

where $p_r$ and $p_v$ denote the densities for the respective random variables. Then the marginal over $\mathbf{r}^{(1)}$ can be expressed as follows:

$$
p_r\left(\mathbf{r}^{(1)}\Big|x\right) = \int_{\mathbf{r}^{(2)}} p_r\left(\mathbf{r}^{(1)}, \mathbf{r}^{(2)}\Big|x\right)\, d\mathbf{r}^{(2)}
$$
$$
= \int_{\mathbf{r}^{(2)}} p_v\left(T_1\left(\mathbf{r}^{(1)}\right), T_2\left(\mathbf{r}^{(2)}\right)\Big|x\right) \cdot \left|\det\nabla_{\mathbf{r}^{(1)}} T_1\left(\mathbf{r}^{(1)}\right)\right| \cdot \left|\det\nabla_{\mathbf{r}^{(2)}} T_2\left(\mathbf{r}^{(2)}\right)\right|\, d\mathbf{r}^{(2)}.
$$

We now employ the change of variables with:

$$
\mathbf{r}^{(2)} = T_2^{-1}(\mathbf{v}^{(2)})
$$
$$
\therefore d\mathbf{r}^{(2)} = \left|\det\nabla_{\mathbf{r}^{(2)}} T_2(\mathbf{r}^{(2)})\right|^{-1} d\mathbf{v}^{(2)},
$$

yielding:

$$
p_r\left(\mathbf{r}^{(1)}\Big|x\right) = \int_{\mathbf{v}^{(2)}} p_v\left(T_1\left(\mathbf{r}^{(1)}\right), \mathbf{v}^{(2)}\Big|x\right) \cdot \left|\det\nabla_{\mathbf{r}^{(1)}} T_1\left(\mathbf{r}^{(1)}\right)\right|\, d\mathbf{v}^{(2)}
$$
$$
= \left|\det\nabla_{\mathbf{r}^{(1)}} T_1\left(\mathbf{r}^{(1)}\right)\right| \cdot \int_{\mathbf{v}^{(2)}} p_v\left(T_1\left(\mathbf{r}^{(1)}\right), \mathbf{v}^{(2)}\Big|x\right)\, d\mathbf{v}^{(2)}
$$
$$
= \left|\det\nabla_{\mathbf{r}^{(1)}} T_1\left(\mathbf{r}^{(1)}\right)\right| \cdot p_v\left(T_1\left(\mathbf{r}^{(1)}\right)\Big|x\right)
$$

Hence, the marginal over $\mathbf{r}^{(1)}$ can be simply expressed in terms of marginal distribution over the transformed variable $T_1(\mathbf{r}^{(1)})$. Finally, we can write the conditional distribution over original responses in terms of the conditionals over the transformed variables:

$$
p_r\left(\mathbf{r}^{(1)}\Big|\mathbf{r}^{(2)}, x\right) = \frac{p_r\left(\mathbf{r}^{(1)}, \mathbf{r}^{(2)}\Big|x\right)}{p_r\left(\mathbf{r}^{(2)}\Big|x\right)}
$$
$$
= \frac{p_v\left(T_1\left(\mathbf{r}^{(1)}\right), T_2\left(\mathbf{r}^{(2)}\right)\Big|x\right) \cdot \left|\det\nabla_{\mathbf{r}^{(1)}} T_1\left(\mathbf{r}^{(1)}\right)\right| \cdot \left|\det\nabla_{\mathbf{r}^{(2)}} T_2\left(\mathbf{r}^{(2)}\right)\right|}{\left|\det\nabla_{\mathbf{r}^{(2)}} T_2\left(\mathbf{r}^{(2)}\right)\right| \cdot p_v\left(T_2\left(\mathbf{r}^{(2)}\right)\Big|x\right)}
$$
$$
= \left|\det\nabla_{\mathbf{r}^{(1)}} T_1\left(\mathbf{r}^{(1)}\right)\right| \frac{p_v\left(T_1\left(\mathbf{r}^{(1)}\right), T_2\left(\mathbf{r}^{(2)}\right)\Big|x\right)}{p_v\left(T_2\left(\mathbf{r}^{(2)}\right)\Big|x\right)}
$$
$$
= \left|\det\nabla_{\mathbf{r}^{(1)}} T_1\left(\mathbf{r}^{(1)}\right)\right| p_v\left(T_1\left(\mathbf{r}^{(1)}\right)\Big|T_2\left(\mathbf{r}^{(2)}\right), x\right).
$$

Note again that in order for the expressions for the conditionals and marginals to cleanly reduce, it is essential that the transformation $T(\cdot)$ is separable over the two groups of neurons.

# B   Zero-Inflated Flow-based Factor Analysis (ZIFFA)

**Joint distribution**   Here, we provide the derivation of the joint distribution $p(\mathbf{r}|\mathbf{x})$ of the ZIFFA model. Let $\mathbf{m} \in \{0,1\}^n$ denote whether a neuron has a response $r_i$ below or above the threshold $\rho$ as indicated by $m_i = 0$ or $m_i = 1$, respectively. For a given assignment of $\mathbf{m}$, we model the density of a response vector $\mathbf{r} \in \mathbb{R}^n_{\geq 0}$ as a product of (1) a uniform distribution between 0 and threshold $\rho$ and (2) a joint FlowFA model for above threshold responses. Accordingly, the conditional distribution can be expressed as follows:

$$p(\mathbf{r}|\mathbf{x},\mathbf{m}) = \underbrace{\left( \prod_{\{i:m_i=0\}} [\![ 0 \leq r_i \leq \rho ]\!] \cdot \rho^{-1} \right)}_{\text{Uniform part for all } r_i \text{ with } m_i=0} \cdot$$

$$\underbrace{\left( \prod_{\{i:m_i=1\}} [\![ \rho < r_i ]\!] \right) \cdot \mathcal{N}(T_\phi(\mathbf{r}_+); f_{\theta,+}(\mathbf{x}), \mathbf{C}_+\mathbf{C}_+^\top + \Psi_+) \cdot |\nabla T_\phi(\mathbf{r}_+)|}_{\text{FlowFA part for all } r_i \text{ with } m_i=1},$$

where $\mathbf{r}_+$ and $f_{\theta,+}(\mathbf{x})$ are the sub-vectors corresponding to responses that are above the threshold. Also, $\mathbf{C}_+$ and $\Psi_+$ are sub-matrices of $\mathbf{C}$ and $\Psi$, respectively, only containing entries corresponding to the neurons with above threshold response. We choose $T_\phi$ such that $T_\phi^{-1}(\mathbf{v}) > \rho$, where $\mathbf{v} = T_\phi(\mathbf{r})$. We use a slight abuse of notation and determine the size of $T_\phi(\mathbf{r}_+)$ by the dimensionality of its input $\mathbf{r}_+$. Here $[\![ A ]\!]$ denotes the indicator function for the set $A$. Note that (1) this is a proper density on $\mathbb{R}^n_{\geq 0}$ since it remains non-negative and integrates to one, and that (2) all population responses $\mathbf{r}$ that do not agree with $\mathbf{m}$ (i.e. $m_i = 0$ and $r_i > \rho$, and vice versa) have zero density since one of the indicator functions in the product will be zero (i.e. they enforce $\mathbf{m}$). To get $p(\mathbf{r}|\mathbf{x})$, we marginalize out $\mathbf{m}$. To this end, we model the probability of each $m_i$ independently as a function $q_i(\mathbf{x})$ of the image $\mathbf{x}$. This yields

$$p(\mathbf{m}|\mathbf{x}) = \prod_{i=1}^n q_i(\mathbf{x})^{m_i} (1 - q_i(\mathbf{x}))^{1-m_i},$$

and

$$p(\mathbf{r}|\mathbf{x}) = \sum_{\mathbf{m}\in\{0,1\}^n} p(\mathbf{r}|\mathbf{x},\mathbf{m}) \cdot p(\mathbf{m}|\mathbf{x})$$

$$= \left( \prod_{\{i:r_i\leq\rho\}} \frac{1 - q_i(\mathbf{x})}{\rho} \right) \cdot$$

$$\left( \prod_{\{i:r_i>\rho\}} q_i(\mathbf{x}) \right) \cdot \mathcal{N}(T_\phi(\mathbf{r}_+); f_{\theta,+}(\mathbf{x}), \mathbf{C}_+\mathbf{C}_+^\top + \Psi_+) \cdot |\nabla T_\phi(\mathbf{r}_+)|.$$

Note that all $2^n - 1$ mixture components whose $\mathbf{m}$ are not in agreement with $\mathbf{r}$ are zero, which leaves only one single mixture component in the end.

**Conditional distribution**   The conditional distribution over $i^{\text{th}}$ neuron's response $r_i$ given the response of all other neurons $\mathbf{r}_{\backslash i}$, can be computed as:

$$p(r_i \mid \mathbf{r}_{\backslash i}, \mathbf{x}) = \frac{p(\mathbf{r} \mid \mathbf{x})}{p(\mathbf{r}_{\backslash i} \mid \mathbf{x})}$$

$$= \begin{cases} (1 - q_i(\mathbf{x})) \cdot \rho^{-1} & \text{if } r_i \leq \rho \\ q_i(\mathbf{x}) \cdot \dfrac{\mathcal{N}(T_\phi(\mathbf{r}_+); f_{\theta,+}(\mathbf{x}), \mathbf{C}_+\mathbf{C}_+^\top + \Psi_+) \cdot |\nabla T_\phi(\mathbf{r}_+)|}{\mathcal{N}(T_\phi(\mathbf{r}_{+\backslash i}); f_{\theta,+\backslash i}(\mathbf{x}), \mathbf{C}_{+\backslash i}\mathbf{C}_{+\backslash i}^\top + \Psi_{+\backslash i}) \cdot |\nabla T_\phi(\mathbf{r}_{+\backslash i})|} & \text{if } r_i > \rho, \end{cases}$$

where subscript $+ \setminus i$ is used to denote all neurons with responses above threshold except for the $i^{\text{th}}$ neuron. While conditioning does not change the distribution over the responses below the threshold $\rho$, for the responses above the threshold, the conditional distribution is computed as the fraction of joint distribution of all neurons $p(\mathbf{r}|\mathbf{x})$ over the joint distribution of all neurons except the target neuron $p(\mathbf{r}_{\setminus i}, \mathbf{x})$. This fraction of the two Gaussian distributions is equivalent to a Gaussian distribution over the response of the target neuron $i$ where the mean and variance are computed conditioned on other neurons $\setminus i$:

$$\frac{\mathcal{N}(T_\phi(\mathbf{r}_+); f_{\theta,+}(\mathbf{x}), \mathbf{C}_+ \mathbf{C}_+^\top + \Psi_+)}{\mathcal{N}(T_\phi(\mathbf{r}_{+\setminus i}); f_{\theta,+\setminus i}(\mathbf{x}), \mathbf{C}_{+\setminus i} \mathbf{C}_{+\setminus i}^\top + \Psi_{+\setminus i})} = \mathcal{N}(T_\phi(r_i); \mu_i, \sigma_i^2),$$

where $\mu_i$ and $\sigma_i^2$ are the posterior mean and variance, respectively, of the $i^{\text{th}}$ neuron's transformed response conditioned on the stimulus $\mathbf{x}$ and transformed responses of other neurons $T_\phi(\mathbf{r}_{+\setminus i})$. These quantities can be straightforwardly computed from the FA model as follows:

$$\mu_i = f_{\theta,+,i}(\mathbf{x}) + \Sigma_{+,i,\setminus i} \Sigma_{+,\setminus i,\setminus i}^{-1}(T_\phi(\mathbf{r}_{+\setminus i}) - \mathbf{f}_{\theta,+,\setminus i}(\mathbf{x}))$$

$$\sigma_i^2 = \Sigma_{+,i,i} + \Sigma_{+,i,\setminus i} \Sigma_{+,\setminus i,\setminus i}^{-1} \Sigma_{+,i,\setminus i}^\top,$$

where $\Sigma = \mathbf{C}\mathbf{C}^\top + \Psi$ and $\Sigma_+ = \mathbf{C}_+ \mathbf{C}_+^\top + \Psi_+$.

It is worth noting that the expressions for the conditionals cleanly reduce only when $T_\phi$ is separable for each neuron (see appendix A for derivations).

## C   Details on data recording and stimulation

Imaging was performed at approximately 9.7Hz for scan 1 and 7.2Hz for scan 2. The recorded visual areas were identified based on retinotopic maps generated as previously described [49, 58]. We selected cells based on a classifier for somata on the segmented cell masks and deconvolved their fluorescence traces using the CNMF algorithm [59].

Images were presented for 500 ms followed by a blank screen with a random duration uniformly distributed between 300 and 500 ms. After spike inference from Calcium data, the neural responses were extracted as the accumulated activity of each neuron between 50 and 550 ms after stimulus onset. All behavior traces (i.e. pupil dilation and running speed) were extracted using the same temporal offset and integration window. The neural responses traces were normalized by their standard deviation computed on the training set.

## D   Additional details about model training

The models were trained end-to-end via gradient-based optimization to maximize the $\log$-likelihood obtained from Eq. (1), (2), (3) or (4) for the corresponding model, optimizing over all parameters of the model. For optimization, we used Adam [60] with (i) an early stopping mechanism [61] that would stop the training if the log-likelihood does not improve for twenty training iterations, and (ii) a learning rate scheduler that reduces the learning rate by a factor of 0.3 if the log-likelihood does not improve for ten training iterations.

To find the best image-computable model, we used Bayesian optimization [48] to find hyper-parameters that optimized the final log-likelihood (explained in section 2.4) of the trained model. Hyper-parameters included the learning rate and the regularization coefficient on the readout weights. The ZIFFA and ZIG models included the zero-threshold parameter $\rho$ as an additional hyper-parameter. To find $\rho$, we experimented with several candidate values and chose the value which resulted in the highest score for the ZIG model, and used the same value for the ZIFFA model.

Each instance of the model with a specific choice of hyper-parameters was trained on a workstation with a single NVIDIA GeForce RTX 2080 Ti GPU. A single ZIFFA model takes approximately 2–3 hours to train whereas all other models take approximately 20–30 minutes to train. The hyperparameter search was completed using one GPU for a total of ~20 hours. All code for model definition, training, and evaluation were implemented in Python 3.8 using PyTorch [62] and NumPy [63] packages.

# E   Computation of conditional response predictions

We estimated the posterior mean of the neuron's responses to an image $\mathbf{x}$ conditioned on the responses of other neurons via Monte Carlo approximation. To achieve this, we first drew samples from the posterior based on the learned FA model, yielding samples in the space of the transformed responses. We then inverse-transformed these samples to yield samples in the space of the neural responses. Subsequently, we computed the average across these samples.

More specifically, for the FA-based models (except ZIFFA, see below), the posterior mean of the neuron's original response to image $\mathbf{x}$ was computed as $\mathbb{E}[r_i|\mathbf{x}, \mathbf{r}_{\setminus i}] = \frac{1}{N} \sum_j^N T_{\phi,i}^{-1}(\mathbf{s}_i^{(j)})$ where $\mathbf{s}_i^{(j)} \sim \mathcal{N}(\mathbb{E}[v_i|\mathbf{x}, \mathbf{v}_{\setminus i}], \sigma_i^2)$. $\mathbb{E}[v_i|\mathbf{x}, \mathbf{v}_{\setminus i}]$ and $\sigma_i^2$ are the posterior mean and variance, respectively, of the $i^{\text{th}}$ neuron's transformed response conditioned on the stimulus $\mathbf{x}$ and transformed responses of other neurons $\mathbf{v}_{\setminus i} = T_\phi(\mathbf{r}_{\setminus i})$. These quantities can be straightforwardly computed from the FA model as follows:

$$\mathbb{E}[v_i|\mathbf{x}, \mathbf{v}_{\setminus i}] = f_{\theta,i}(\mathbf{x}) + \Sigma_{i,\setminus i}\mathbf{\Sigma}_{\setminus i,\setminus i}^{-1}(T_\phi(\mathbf{r}_{\setminus i}) - \mathbf{f}_{\theta,\setminus i}(\mathbf{x})),$$
$$\sigma_i^2 = \Sigma_{i,i} + \Sigma_{i,\setminus i}\mathbf{\Sigma}_{\setminus i,\setminus i}^{-1}\Sigma_{i,\setminus i}^\top,$$

where $\mathbf{\Sigma} = \mathbf{C}\mathbf{C}^\top + \Psi$.

For the ZIFFA model, the procedure for posterior mean computation is almost identical to the procedure explained above with two differences: 1) when computing the posterior mean and variance of the neuron's transformed response, we condition only on other neurons who exhibit above threshold responses $\mathbf{r}_{+\setminus i}$ (refer to appendix B for details), and 2) the posterior mean in the neural response space is computed as the mixture of the mean of the two mixture model components:

$$\mathbb{E}[r_i|\mathbf{x}, \mathbf{r}_{\setminus i}] = (1 - q_i(\mathbf{x})) \cdot \frac{\rho}{2} + q_i(\mathbf{x}) \cdot \mathbb{E}[r_i|\mathbf{x}, \mathbf{r}_{+\setminus i}].$$

# F   Synthetic data generation

We generated 5,000 samples from a correlated 100-d Gaussian distribution, corresponding to the transformed responses $\mathbf{v}$ of 100 neurons. The covariance matrix of the Gaussian distribution took the form $CC^\top + \Psi$, corresponding to that of FA models. $CC^\top$ was of rank 4 with $C \in \mathbb{R}^{100 \times 4}$, where the choice of the rank was arbitrary. To ensure that generated Gaussian samples (1) fall in a range where the transformation is invertible and that they (2) cover the most nonlinear part of the transformation, we kept the variances and covariances relatively small and sampled the mean for each neuron in a transform-specific fashion. The entries of $C$ were sampled uniformly between 0.02 and 0.07, and the diagonal entries of $\Psi$ were sampled uniformly between 0.002 and 0.01. We further imposed stronger or weaker correlations between selected neurons by scaling the corresponding entries of the full covariance matrix either by 1.5 or 0.2. The mean for each neuron (in the transformed response space) was uniformly sampled between a transform-specific minimum and maximum value. The transform-specific minimum value was computed as $T(\epsilon) + \alpha \cdot \max(CC^\top + \Psi)$ where $\epsilon$ was a small value ($10^{-12}$) close to zero and $\alpha$ took on a transform-specific value summarized in Table 1. The transform-specific maximum value was computed as $T(10)$. Once the Gaussian samples were generated for each transformation function, the samples were inverse-transformed via the corresponding $T^{-1}$ into the simulated neural responses. The code used to generate simulated data can be found at `https://github.com/sinzlab/bashiri-et-al-2021`.

Table 1: transform-specific $\alpha$ values

| $T$: | identity | sqrt | anscombe | example 1 | example 2 | example 3 | example 4 |
|---|---|---|---|---|---|---|---|
| $\alpha$: | 1.0 | 3.0 | 2.0 | 1.5 | 3.0 | 3.0 | 1.0 |

| $T$: | example5 | example 6 | example 7 | example 8 | example 9 | example 10 | |
|---|---|---|---|---|---|---|---|
| $\alpha$: | 3.0 | 3.0 | 3.0 | 1.0 | 3.0 | 3.0 | |

## G Computing orthonormalized latent states

We extract latent states from the FA-based model by computing the posterior mean $\mathbb{E}[\mathbf{z}|\mathbf{x}, \mathbf{r}]$. While the relationship between the latent states $\mathbf{z}$ and the neural responses $\mathbf{r}$ is well defined via the model relationship $\mathbf{r} = T_\phi^{-1}(\mathbf{f}_\theta(\mathbf{x}) + \mathbf{C}\mathbf{z} + \epsilon)$, the factor loading matrix $\mathbf{C}$ can only be uniquely determined up to an arbitrary orthogonal transformation. That is, given $\mathbf{z} \sim \mathcal{N}(0, I_k)$, we can transform the factor loading matrix $\mathbf{C}$ and $\mathbf{z}$ by any arbitrary orthogonal transform matrix $\mathbf{R}$ to yield $\mathbf{C}' = \mathbf{C}\mathbf{R}$ and $\mathbf{z}' = \mathbf{R}^\top \mathbf{z}$. The resultant alternative definition of $\mathbf{z}'$ along with $\mathbf{C}'$ would yield identical fit to the neural responses since $\mathbf{C}'\mathbf{z}' = \mathbf{C}\mathbf{R}\mathbf{R}^\top \mathbf{z} = \mathbf{C}\mathbf{z}$ and $\mathbf{z}' \sim \mathcal{N}(0, I_k)$. Furthermore, the inferred latent states $\mathbf{z}$ are not necessarily ordered by how much neural variability they account for. In fact, the order of the latent states are arbitrary, and this can be seen by noting that a permutation matrix is an example of an orthogonal transformation. Combined with an additional observation that the columns of $\mathbf{C}$ are not guaranteed to be mutually orthogonal, interpreting the inferred latent states $\mathbf{z}$ is difficult and quite arbitrary.

To address this issue, we follow a similar approach to Yu et al. [30]. Briefly, we orthonormalize the columns of $\mathbf{C}$ by applying the singular value decomposition to the learned $\mathbf{C}$ which yields $\mathbf{C} = \mathbf{U}\mathbf{D}\mathbf{V}^\top$. As a result, $\mathbf{C}\mathbf{z}$ can be re-written as $\mathbf{C}\mathbf{z} = \mathbf{U}(\mathbf{D}\mathbf{V}^\top \mathbf{z}) = \mathbf{U}\tilde{\mathbf{z}}$ where $\tilde{\mathbf{z}} \equiv \mathbf{D}\mathbf{V}^\top \mathbf{z}$ is the *orthonormalized latent state*. Consequently, instead of visualizing the MAP of $\mathbf{z}$, $\mathbb{E}[\mathbf{z}|\mathbf{x}, \mathbf{r}]$, we would visualize $\mathbf{D}\mathbf{V}^\top \mathbb{E}[\mathbf{z}|\mathbf{x}, \mathbf{r}]$. This approach incurs multiple advantages. Firstly, while the elements of $\mathbf{z}$ (and corresponding columns of $\mathbf{C}$) have no particular order, the elements of $\tilde{\mathbf{z}}$ (and corresponding columns of $\mathbf{U}$) are ordered by the amount of data variance they explain. Therefore, the inferred latent states are ordered by their contribution in explaining the variance observed in neural activity, resulting in more intuitive and interpretable latent states. Secondly, when the singular values are non-zero and non-repeating, the method recovers a unique latent state $\tilde{\mathbf{z}}$ for $\mathbf{C}' \equiv \mathbf{C}\mathbf{R}$ and $\mathbf{z}' \equiv \mathbf{R}^\top \mathbf{z}$ regardless of $\mathbf{R}$. This can be seen from the fact that singular value decomposition of $\mathbf{C}'$ is given by $\mathbf{C}' = \mathbf{U}\mathbf{D}\mathbf{V}'^\top$ where $\mathbf{V}' = \mathbf{R}^\top \mathbf{V}$, therefore

$$
\begin{aligned}
\tilde{\mathbf{z}}' &\equiv \mathbf{D}\mathbf{V}'^\top \mathbf{z}' \\
&= \mathbf{D}\mathbf{V}^\top \mathbf{R}\mathbf{R}^\top \mathbf{z} \\
&= \mathbf{D}\mathbf{V}^\top \mathbf{z} \\
&= \tilde{\mathbf{z}}.
\end{aligned}
$$

# H  Supplementary Figures

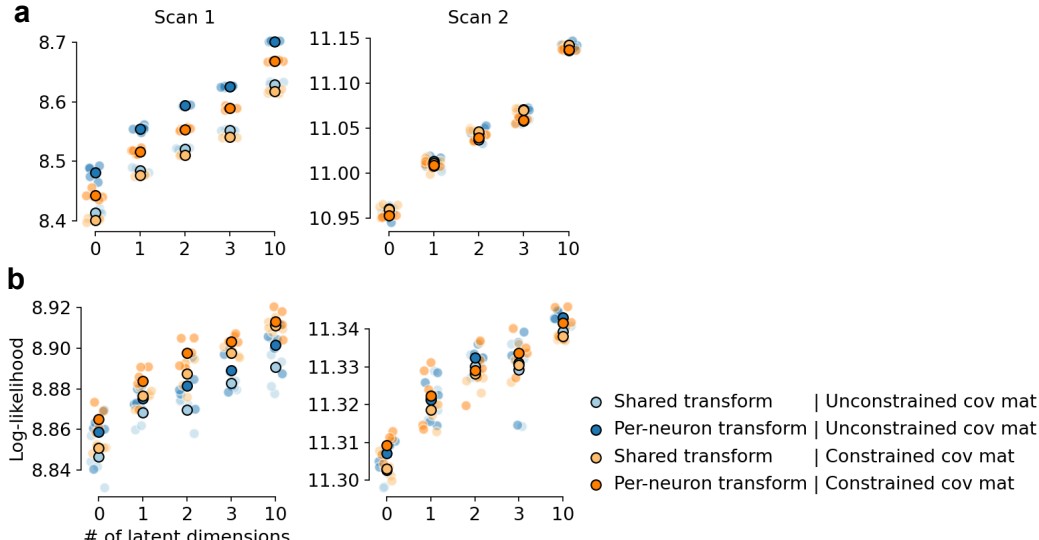

Figure S1: Comparison of flow-based models with different model configurations. These configurations include: 1) using a shared vs neuron-specific flow transformation, and 2) unconstrained vs constrained covariance matrix of the FA. The transformation $T_\phi$ could be defined such that a single flow transformation is shared among all neurons or it could be defined such that it contains neuron-specific parameters resulting in neuron-specific transformations (for details refer to section 2.2). As expected, per-neuron transformation (darker color) seem to results in a higher likelihood. The constrain imposed on the covariance matrix was used to ensure that the marginals have unit variance (i.e. a correlation matrix). While unconstrained covariance matrix (blue color) works best for the FlowFA model, the ZIFFA model with constrained covariance matrix (orange color) generally results in highest likelihood. **a**: FlowFA model. **b**: ZIFFA model.

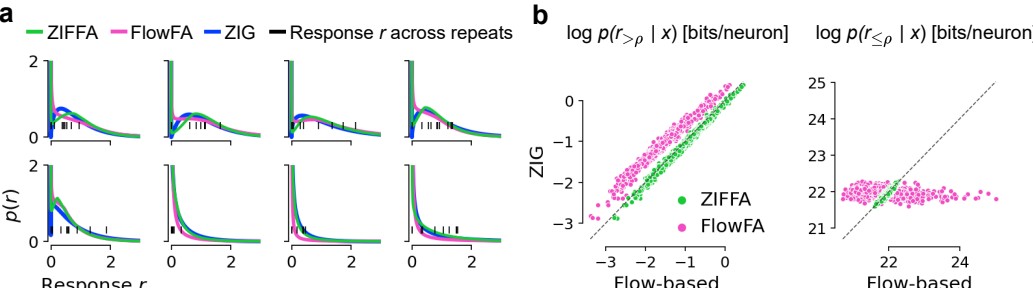

Figure S2: Comparison of the learned density by the ZIFFA, FlowFA, and ZIG models. **a**: Example marginal distribution of responses of 8 sample neurons to the repeated presentations of an image from the test set and the corresponding fits of ZIFFA, FlowFA, and ZIG. While all three models peak at zero, the FlowFA puts relatively little probability mass on positive responses $\mathbf{r}_{>\rho} = \{r_i | r_i(\mathbf{x}) > \rho\}$. **b**: Flow-based models vs ZIG log-likelihood in bits/neuron for positive responses $\mathbf{r}_{>\rho}$ and "zero" responses $\mathbf{r}_{\leq\rho}$, respectively. Each point is a single trial. Compared to ZIFFA and ZIG, FlowFA model seems to put less mass on responses $\mathbf{r}_{>\rho}$ and, for many trials, more mass on responses $\mathbf{r}_{\leq\rho}$. Importantly, while ZIFFA performs very similar to ZIG for responses $\mathbf{r}_{\leq\rho}$, it slightly puts more mass on the responses $\mathbf{r}_{>\rho}$ resulting in a higher likelihood performance as illustrated in (Fig. 3).

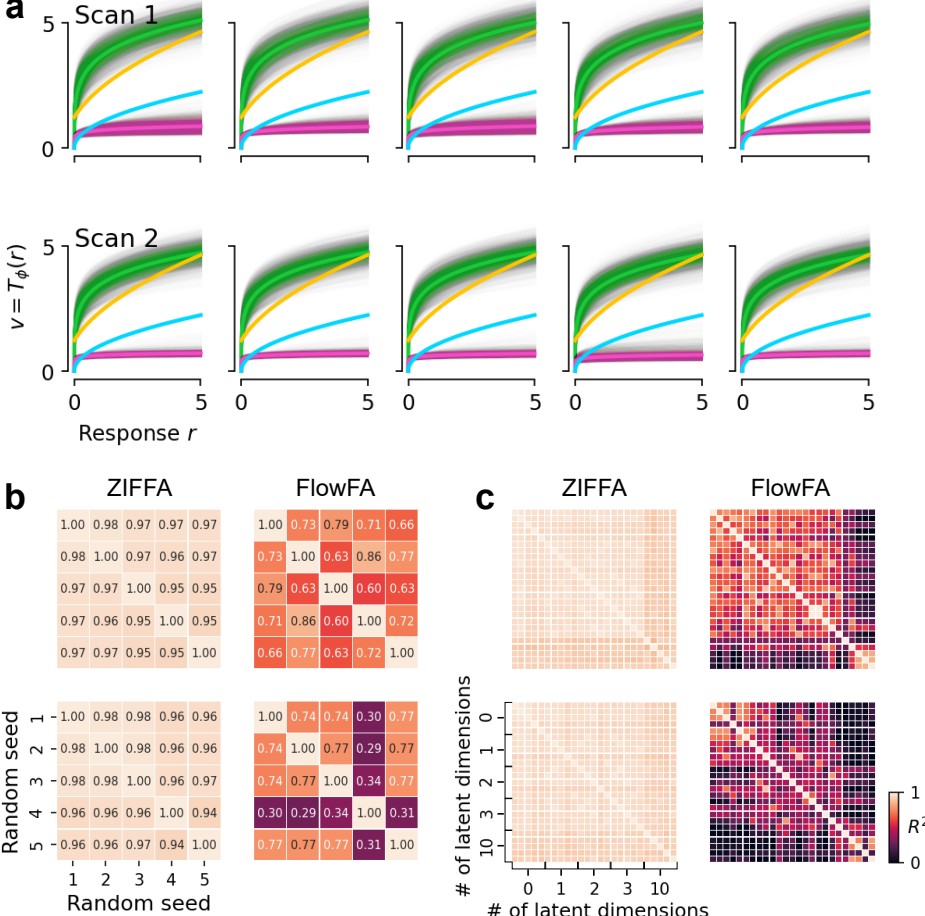

Figure S3: Consistency of the learned transformation across models initialized and trained with different random seeds, and across different number of latent dimensions. **a**: The learned flow transformation for both ZIFFA (green) and FlowFA (pink) models with 0-dimensional latent. Square-root (blue) and Anscombe (yellow) are also visualized for reference. Top row: Scan 1; bottom row: Scan 2. Colors are the same as in Fig. 3. **b**: Quantification of the consistency of learned flow transformations across random seeds, for the same models shown in **a**. To quantify the consistency, we flattened "transformed" responses **v** across all neurons getting a single vector for one seed, and then computed the $R^2$ between flattened **v** of all pairs of seeds. Higher $R^2$ value implies more consistency. Top row: Scan 1; bottom row: Scan 2; Left column: ZIFFA; right column: FlowFA. **c**: Same as **b**, but extended to also show the consistency of the learned transformation across models with different number of latent dimensions.

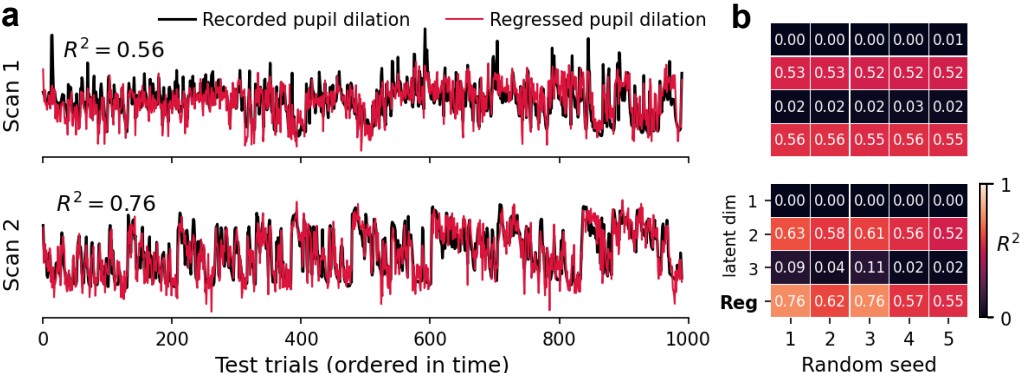

Figure S4: Correlation and regression analysis between inferred latent states and the pupil dilation. **a**: The regressed pupil dilation vs the recorded pupil dilation for the same model as in Fig. 4. **b**: First three rows: The $R^2$ values between orthonormalized latent states and pupil dilation across all random seeds. Last row: The $R^2$ values between regressed and recorded pupil dilation. Top: scan 1; bottom: scan 2.