# OpenReview forum: "A flow-based latent state generative model of neural population responses to natural images"
_NeurIPS.cc/2021/Conference — NeurIPS 2021 Spotlight_

### Official Review · Reviewer_Two8 · 2021-07-08

**Rating:** 7
**Confidence:** 3

**Summary:**

The authors propose an algorithm based on a combination of a DNN-based model (that account for stimulus-variability) and a flow-based model (accounting for inherent neural variability) to model the behavior of neural responses. In this paper, the algorithm, called FA-flow (compared with ZIG, Poisson and ZA-Fixed) is trained on pairs of image/neural recording from bi-photon microscope on 2 mices. The authors claim is that their algorithm perform similarly to current state of the model neural response while additionally yielding latent states that exhibit meaningful relations to anatomical and functional properties.

**Limitations And Societal Impact:**

The authors have not addressed the societal impact of their work

**Main Review:**

## General comments :
This article is in general interesting, and bring new Machine Learning concepts (i.e. the normalizing Flow) into computational Neuroscience. The main claim of the article, that is adding a flow on top of a DNN-based model allow to model both types of neural variability (i.e. stimuli variability and inherent neural variability) is not properly demonstrated. Nevertheless, the experiments conducted by the authors demonstrate the FA-Flow algorithm produce a relatively good fit of neural responses, but some of the conclusion made by the authors are not support by quantitative results (i.e. RFs mappings) or suffer from low statistical significance (see below for more details). In addition, some of the conclusion made by the authors are complicated to grasp (especially the conclusion from the Fig 4b and c).  My score : 5.

## More specifics comments :
Section Methods : This section is clear and well written. Minor question : The authors are using 1000 V1 neurons to train the model on scan 1, and they report a total number of V1 neurons of 2867. For scan 2 the selection process is described (this is random) but  not for scan 1. Could please describe the neuron’s selection process for V1 neurons on scan 1 ?

### Section Results :
#### Section 3.1 :
I think a crucial experiment is missing in this section. One of the main claim of the authors is that flexible latent model better capture intrinsic correlated neural activity, and this claim is not demonstrated for their FA-Flow model. On my understanding the authors have all the datas to conduct such an experiment : some of the images are repeated 10 times (see line 172), thus one can see on these specific images how the FA-Flow model is fitting the intrinsic neural variability compared to the other models. Such an experiments would support the main claim of the article.

Figure 3 : Line 236, the authors state : « with Increasing latent dimension the FlowFA model … approaches the ZIG model in terms of likelihood’.  I do not see this improvement of log-likelihood with the increase of latent dimensions. As this is far from being obvious, the authors should conduct statistical test to support their claim.

Figure 4a : We see qualitatively that the FAFlow algorithm allows deducing a relatively good mapping between the RF position in the image space and neurons position in the cortex. But without further quantification of the quality of the mapping, it is hard to draw a solid conclusion. Is it possible to quantify the quality of the mapping ?

Figure 4b and c : I have trouble understanding the claim made by the authors : ’the effect of some latent states vary systematically accros brain area, as a function of RF location' (line 255). Could you please further explain how you are drawing this conclusion. Also a legend showing the meaning of the colors would help to better understand these 2 subfigures.

Figure 4d : Without further comparison it is complicated to say that ’some of the inferred latent state correlated strongly with pupil dilatation' (see line 259). For example in scan 1, the latent 3 accounts for only 41% of the variability (r2=0.41) which is usually not high enough to consider a good fit. Further statistical tests need to be conducted to be able to conclude on the significance of the prediction (and ideally a comparison with other r2 given in the literature).


**Time Spent Reviewing:**

4h

---

> ### Author Response · Authors · 2021-08-10
> **Authors response to Reviewer Two8**
>
> Thank you for your comments and feedback. We performed additional analyses to assess noise correlation and the statistical significance of our observations/claims in the manuscript.
>
>
> - **Re: Selection of neurons for Scan 1**: The neurons were randomly selected for both scans. We will make sure this is clearly stated in the text.
>
>
> - **Re: Additional experiments regarding noise correlations**: We observed that increasing the number of latent dimensions lead to the increase in model’s predictive performance as measured by likelihood of the responses (see below for statistical analysis of this), and that the latent variables highly correlate with pupil dilation, a known surrogate for global brain states. This already suggests that the model is improving because it is better able to capture non-trivial correlation structure among neurons (especially as we move from 0 latent - independence - to >0 latents with modeled dependencies). However, it’s not clear whether there are any stimulus dependent noise correlations and whether they can be picked up by the model.
>
>     &nbsp;&nbsp;&nbsp;&nbsp; To this end, we compared the noise correlations computed per image across 10 repeated presentations on our test set with the noise correlations of a shuffled dataset where we destroyed the stimulus-dependency via shuffling across stimuli. Specifically, we  randomly shuffled the trial IDs across paired neuronal responses. Since we shuffled the pairs across trials, all common “brain state” correlations should still be intact, while any image dependent noise correlations should be destroyed. We observed that none of the neuron pairs exhibit statistically significant differences between the shuffled and unshuffled noise correlations (P > 0.05 Wilcoxon signed-rank test, Bonferroni corrected for multiple comparisons), for both scans. This suggests that noise correlations in our dataset do not show observable dependence on the stimuli. Therefore, there is no meaningful dependence of the noise correlations on the stimuli in the data that could be captured by our model. This could be a result from computing noise correlations with only 10 repeats per image, which is a much lower number of repeats than commonly used in literature and therefore we expect the computed noise correlations to be rather noisy. In the final version of the manuscript, we will mention this analysis. We would like to leave for future work to compare between the current ZIFFA model and an extension that also changes noise correlations as a function of the stimulus (by making C image dependent) on the appropriate dataset (that includes such noise correlations).
>
>
> - **Re: Improvement of log-likelihood with increasing number of latent dimensions**: Our claim about the increase in FlowFA performance with increasing latent dimensions is based on observing a trend in the results (Fig 2a insets). However, since the time of submission we have improved the model (ZIFFA; see general response) and the optimization process and the trend is much more clear now. More specifically, with increasing latent dimensions the performance of the FlowFA improves as follows: 0-, 1-, 2-, and 3-dimensional latent state models result in log-likelihood values of 8.48, 8.55, 8.59, and 8.63 for scan 1 and 10.95, 11.01, 11.04, 11.07 for scan 2. For both scans, the improvement in log-likelihood is statistically significant with P < 0.01 (Wilcoxon rank-sum test) for the FlowFA model across all latent dimensions. Similarly, the log-likelihood of the ZIFFA model improves across latent dimensions: 0-, 1-, 2-, and 3-dimensional latent state models result in log-likelihood values of 8.87, 8.88, 8.89, and 8.90 for scan 1, and 11.31, 11.322, 11.329, and 11.334 for scan 2. However, while a trend can be observed, the improvement in log-likelihood with increasing number of latent dimensions is not statistically significant for the ZIFFA model. We will include the quantification of statistical significance of the improvement in the model performance with increasing number of latent dimensions in the final revision of the manuscript.
>
>
> - **Re: Further quantification of the quality of the mapping between cortex positions and RF positions**: The quality of the learned mapping is demonstrated by the comparison between experimentally identified brain areas shown in panel *Cortex Positions* in Fig 4a (each area illustrated with a different color) vs the brain areas identified via our model shown in panel *det(Jacobian)* in Fig 4a. Importantly, the experimental procedure for identifying brain areas were independent of the data (both images and responses) used for training the model. Therefore, the fact that our model which is trained on responses of neurons to natural images can segregate areas and also correctly quantify flips (red vs blue color) of the visual space with respect to the cortex positions shows the utility of the model in yielding experimentally verified functional insights about the visual sensory neurons. To further assess the quality of the mapping between cortex positions and RF positions, we quantified how well our model can classify distinct brain areas. The result shows that our best performing ZIFFA model with 3 latent dimensions correctly classifies distinct brain areas with accuracy of 94% and 97% for scan 1 and scan 2, respectively. Across models initialized and trained with different random seeds, on average the ZIFFA model correctly classifies distinct brain areas with accuracy of 84% (SEM = 0.04) and 75% (SEM = 0.08). It is important to note that the experimentally identified brain areas cannot be taken as the absolute ground truth since the experimental procedure to determine the areas is quite coarse (please refer to our response to **Re: Does the inclusion of correlated variability in the model improve the inferred estimates of $\delta$ compared to a model with C=0? of reviewer BNcm)**. Moreover, also based on the comments we received from the other reviewers, we see that the description we provide in the current version of the manuscript does not seem to clearly explain and highlight the significance and the implications of the results related to our model-based area identification. We will ensure that the explanation is more detailed, clear, and understandable in the final version.
>
>
> - **Re: How the claim “the effect of some latent states vary systematically across brain areas, as a function of RF location” was made**: With this analysis, our goal was to see whether the weights that project latent states onto the neurons (captured by the factor loading matrix $C$) have a structure when plotted on the cortical positions and RF positions. The color therefore shows the weight strength for each latent dimension across all neurons, either on their cortical positions (Fig 4b) or on their RF positions (Fig 4c).  Our goal here was to show-case the kinds of biologically-relevant analysis that can be performed via the presented model. However, to draw biological conclusions, more controls are necessary which goes beyond the scope (and the intention) of the current paper. Concerns about the results/interpretations of Fig 4 were also raised by other reviewers, and we will make sure to clearly state the purpose of the figure and also point out other possible interpretations of the results shown in the figure.
>
>
> - **Re: Statistical significance of the correlation between inferred latent states and pupil dilation**: Pupil dilation is a known correlate of brain state [1, 2]. Our goal with this analysis (Fig 4d) was to assess whether our model is able to extract relevant brain state signals. The correlation values between the pupil dilation and the inferred latent states from our model are indeed similar, or even stronger, than the values previously reported in the literature. More specifically, our latest model (ZIFFA) with 3-dimensional latent state trained on scan 1 yields $R^2$ values of 0.33, 0.25, and 0.08 (P < 0.001) between each latent state, respectively, and pupil dilation. Similarly, for scan 2 the model yields $R^2$ values of 0.09, 0.15, and 0.55 (P < 0.001). $R^2$ values reported in the literature generally fall within the range of 0 to 0.6 (supplementary Fig 4c in ref 1, and supplementary Fig 8a in ref 2). To further assess the relevance of the inferred latent states, we performed a regression analysis to assess how well we can predict pupil dilation when combining all three latent states. The result of this analysis shows that the $R^2$ value between the pupil dilation and the regressed output is 0.56 (P < 0.001) and 0.76 (P < 0.001) for scan 1 and scan 2, respectively. In the final version of the manuscript, we will 1) include the regression analysis to show how well pupil dilation can be regressed onto the inferred latent states, and 2) compare the reported $R^2$ values with the previously reported results in the literature.
>
>
> [1] Stringer, Carsen, et al. "Spontaneous behaviors drive multidimensional, brainwide activity." Science 364.6437 (2019).
>
> [2] Reimer, Jacob, et al. "Pupil fluctuations track rapid changes in adrenergic and cholinergic activity in cortex." Nature communications 7.1 (2016): 1-7.

---

> > ### Comment · Reviewer_Two8 · 2021-08-17
> > **Response to authors**
> >
> > Thanks for the authors for their detail response and the complementary experiments they have run.
> >
> > The authors have in general well responded to my comments (except for the noise correlation which I indeed think it is coming from the small number of repeated stimuli). I am willing to increase my rating from 5 to 7.

---

> > > ### Author Response · Authors · 2021-08-17
> > > **Authors response to Reviewer Two8**
> > >
> > > Thank you very much for your positive feedback and your willingness to increase your score to 7. We are happy to hear that our responses seem to have addressed many of your concerns. Regarding the noise correlations, we completely agree with you that one likely reason for why we cannot detect any meaningful stimulus-dependent noise correlations is the low number of repeats (only 10) per stimulus in our dataset. Unfortunately, we cannot change that for the given dataset. Thus, we will reserve the analysis to when an appropriate dataset is available to us.
> > >
> > > With regards to non-stimulus-dependent noise correlations, we still think that our model picks up meaningful structure in the data as demonstrated by 1) the increase in log-likelihood as the number of latent dimensions increases and 2) the high correlation between inferred latent states and known surrogates for brain state (i.e. pupil dilation).

---

### Official Review · Reviewer_mWE7 · 2021-07-14

**Rating:** 6
**Confidence:** 4

**Summary:**

In the paper "A Flow-based latent state generative model of neural population responses to natural images", the authors propose a model that can take into account stimulus driven stimulus-conditioned neural activity. The model is based on normalizing flows for the marginals and on a factor analysis model for modelling covariances and is validated on simulated data as well as data recorded from mouse visual cortex.

**Limitations And Societal Impact:**

Limitations of the work with respect to the ZIP marginals are discussed to some extend in the Discussion but important points are left out. The FlowFA model assumes a stimulus-independent Gaussian dependence structure which is transformed by marginal stimulus-dependent nonlinearities. How flexible is the resulting dependence structure? Is the flow transformation of line 129 limiting? Such questions are not addressed in the manuscript. Societal implications with respect to animal experiments are briefly mentioned in the Discussion.

**Main Review:**

Originality:
To the best of my knowledge, the factor analysis approach combined with normalizing flows for the marginals is novel. That said, many non-cited models have previously been proposed, explicitly taking stimulus driven and stimulus-conditioned neural activity into account. For instance, Keeley, Aoi, Yu, Smith and Pillow, bioRxiv 2020 explicitly model signal and noise components in a Gaussian process factor analysis framework. Sorochynskyi, Deny, Marre and Ferrari, PLoS Comput Biol 2021 used a maximum entropy model and explicitly take noise correlations into account as well.

Quality:
The flow transformations are compared against two fixed transformations: square-root and Anscombe. The alternatives being fixed, it is not surprising that the flow model outperforms these fixed transformations. A comparison to a more flexible transformation such as a kernel density estimator would have been more insightful. Figure 5 suggests that even the ZIG model does a better job of modelling the marginals of the analyzed dataset.
Similarly, the comparisons to the control models are not fair. Contrary to the proposed model, both the independent Poisson and the ZIG model assume independence (line 137).
Tractable maximum entropy models (e.g. O'Donnel et al. Neural Comput 2017) or the dichotomized Gaussian model with appropriate marginals (Macke et al., Neural Comput 2009) would have been better contenders.

Clarity:
The clarity of the manuscript is fine but could be improved. In line 87, we learn that neural responses are represented as real numbers but it is not clear what these numbers mean until much later in the manuscript. Sometimes the model is called FAFlow (e.g. Fig 5), other times FlowFA (e.g. line 88).

Significance:
Models for taking both stimulus driven and stimulus-conditioned activity into account are important tools in systems neuroscience. Given the lacking comparisons, the value of the proposed model is unclear.

**Time Spent Reviewing:**

4

---

> ### Author Response · Authors · 2021-08-10
> **Authors response to Reviewer mWE7 (Part 2)**
>
> - **Re: Comparing the flow-based model to our control models is not fair because “both the independent Poisson and the ZIG model assume independence”**: We would agree that it would have been unfair to compare our models, specifically those with one or more latent dimensions, to models with independence assumption exclusively on how well they can account for correlation. However, the most important comparison was based on the log-likelihood (logL) which measures how well the modeled distribution matches the target distribution. While our flow model indeed has the advantage of potentially capturing the noise correlation, exactly how much improvement in logL one would obtain by accounting for such correlation was not known. Furthermore, despite their obvious omission of noise correlations, system-identification models based on independent Poisson and ZIG represent the current state-of-the-art predictive models of sensory population responses to natural images. Therefore, these independent models serve as critical comparison targets for our model who aims to improve upon these models by capturing more complex distributions. Finally, note that, when the latent dimension is 0, the FlowFA also has independent marginals and becomes directly comparable to ZIG and Poisson models.
>
>
> - In fact, as the reviewer noted, **“ZIG model does a better job of modeling the marginals,”** and even attained higher logL score compared to the FlowFA models despite not modeling the correlation among the responses. We therefore believe the comparison of our models to these state-of-the-art models in system identification to be necessary and informative. Consequently, the result of this comparison has prompted us to seek an improvement to our FlowFA model, and we have now trained and tested a modification of the model that we call ZIFFA which explicitly tries to capture the zero-inflation in the marginal that biased the fitting of FlowFA models. We provide the description and summary of this in the general response above.
>
>
> - **Re: Lack of information and naming inconsistencies**: Thank you for pointing out the lack of information about the nature of the responses in line 87 and the naming inconsistency. We will ensure to add that the responses are recorded via two-photon Calcium imaging, in line 87. Moreover, we will thoroughly check the paper for typos and naming inconsistencies.

---

> > ### Comment · Reviewer_mWE7 · 2021-08-24
> > **Response to authors**
> >
> > With the clarification of the scope of the work and the performance increase of the new ZIFFA method, the authors convinced me to increase my rating from 4 to 6.

---

> > > ### Author Response · Authors · 2021-08-25
> > > **Thanks**
> > >
> > > Thanks a lot. We're happy to hear that.

---

> ### Author Response · Authors · 2021-08-10
> **Authors response to Reviewer mWE7 (Part 1)**
>
> Thank you very much for your feedback. We appreciate your detailed comments and pointers to additional references. Your comments helped us realize that the submitted version of the manuscript did not clearly highlight what we considered to be the key differences between our approach and the existing methods on modeling noise distributions which we hope to clarify in our response, and in the future revision of the manuscript.
>
>
> - **Re: “[M]any non-cited models have previously been proposed, explicitly taking stimulus driven and stimulus-conditioned neural activity into account” and “[g]iven the lacking comparisons, the value of the proposed model is unclear”**:  We apologize that some of the suggested references (Keeley et al. 2020) were omitted accidentally despite our awareness of the work. We were not aware of the work of Sorochynskyi et.al. 2021, and thank the reviewer for bringing this to our attention. We will make sure to include and discuss all of these citations to the final version of our manuscript.
>
>     &nbsp;&nbsp;&nbsp;&nbsp; While much work exists modeling the stimulus-conditioned noise distributions, we believe that there are crucial differences, specifically in the goals, between our approach and these methods that were not stated clearly enough in the submitted manuscript. We would like to elaborate on this point here and in the future revision.
>
>     &nbsp;&nbsp;&nbsp;&nbsp; The central goal of our methods lies in directly predicting the distribution of neural population responses to each stimulus, denoted as $p(\mathbf{r} | \mathbf{x})$, explicitly **as a function of the stimulus x**. In other words, our goal is to train a predictive model that can predict the distribution of population responses to an entirely novel stimulus with no additional training required. Consequently, we assess our model’s predictive performance on the test set consisting of stimuli (i.e. natural images) that has never been shown as part of the training or validation set. We see our method as a natural extension to existing system-identification models that predict neural responses as a function of stimulus. Many of these are formulated with independent noise assumptions such as Poisson or ZIG, and thus we improve upon this by proposing the FlowFA (or ZIFFA) model that allows us to capture more complex correlated distribution among neurons without losing the end-to-end trainability of the entire model with gradient-based methods, and specifically on a dataset that contains responses to a rich set of stimulus but **without repeats**.
>
>     &nbsp;&nbsp;&nbsp;&nbsp; This approach differs significantly from many of the existing work on modeling noise distributions including the work of Keeley et al. 2020 and Sorochynskyi et al. 2021 - in most of these works, the structure of stimulus conditioned distribution $p(\mathbf{r} | \mathbf{x})$ is fitted separately for each unique stimulus. Furthermore, these methods often explicitly require repeated presentations of the same stimulus for fitting the distribution to the responses. For example, in the case of Keeley et al. 2020, a unique stimulus-specific latent state was learned, and for the case of Sorochynskyi et al. 2021, stimulus-specific CDF was learned for each neuron. While these approaches represent powerful methods to capture the complex distribution $p(\mathbf{r} | \mathbf{x})$, predicting the population distribution to a new stimulus $p(\mathbf{r} | \mathbf{x}^*)$ would require fitting a new set of stimulus-specific parameters. Therefore, existing methods on modeling noise distributions do not readily offer a model that can be used to predict the response distribution to a novel unseen stimulus. This in turn makes it challenging to perform direct comparison to our methods in predicting responses to a novel set of stimuli found in the testset. Furthemore, many of these methods explicitly require the use of responses to repeated stimulus presentations and have no obvious method to combine with the training of the flexible DNN-based neural predictive models. This consequently makes it challenging to offer an extension to the existing approaches that will turn them into a flexible model that would predict response distribution under a new stimulus.
>
>     &nbsp;&nbsp;&nbsp;&nbsp; However, while existing methods present major challenges in creating an end-to-end trainable model for stimulus-conditioned neural distributions that are circumvented in our method, we agree that our flow-based approach is highly related to many of these existing approaches and could have been better highlighted in the manuscript. We will elaborate on these similarities here, and subsequently include this discussion in the revised manuscript:
>
>     &nbsp;&nbsp;&nbsp;&nbsp; Keeley et al. 2020 capture the trial-by-trial fluctuations by modeling the stimulus-specific and trial-specific latents via Factor Analysis (FA) models much like in our model. Importantly, while we capture the dependence of the stimulus-specific latents on the stimulus explicitly via a trained DNN, they inferred it from repeated presentations of the stimulus. Furthermore, the final Poisson distribution used to map from the latents to the distribution of neurons can be captured in our model via the flow-based transformation (e.g. inverse Anscombe) that maps Gaussian-distributed latents into a continuous approximation of a Poisson distribution.
>
>     &nbsp;&nbsp;&nbsp;&nbsp; Our flow-based models can be cast as a generalization of the copula-based distribution approximation proposed by Sorochynskyi et al. 2021 (which in turn is a generalization of the dichotomized Guassian distribution-based methods as noted by the authors). Specifically, the use of FA in our model makes our approach heavily related to Gaussian copula. However, by explicitly limiting the stimulus dependence to occur via the shift in the mean of the FA model, we avoid the reliance on the repeated presentations of the stimuli, which is one drawback of coupla-based methods in predicting responses on our dataset.
>
>     &nbsp;&nbsp;&nbsp;&nbsp; In summary, our model can be seen as a generalization of many previous approaches for modelling stimulus-conditioned neural variability, while being computationally easier to handle at the same time.
>
>
> - **Re: The current discussion of the limitation of FlowFA model is limited and “important points are left out”. (related to previous point)**: We appreciate this comment and agree that indeed we could have elaborated more on the limitation of the model and on the general flow-based modeling approach. We will provide some of the discussion points here, and will elaborate on these points in the revised manuscript. As discussed above, our flow-based approach generalizes several existing methods to capture stimulus-conditioned variability of neural responses, while being more computationally tractable at the same time. This allows us to train it end-to-end directly on the likelihood. Within this general framework, we presented a specific case where we learned **stimulus-independent transformations** that are shared across all neurons, mapping responses into a FA model whose mean varies with the stimulus. As noted earlier, for each stimulus, this approach closely parallels Gaussian copula and thus share much of the same limitations. Also the fact that stimulus-dependence only occurs via the stimulus-based mean of the FA implies that the model can only capture changes in the covariance structure that varies with the mean (although this limitation is shared with many models based on GPFA models with Poisson distribution). That being said, we believe that our FlowFA model represents one of many models to come from the more general framework of flow-based modeling of neural distributions. Relaxing some of the simplifying assumptions we employed in our FlowFA model would allow for the model to immediately overcome some of the above mentioned limitations. For example, one could explore a much more complex and richer set of learnable transformations.
>
>
> - **Re: “It is not surprising that the flow model outperforms these fixed transformations” and a “comparison to a more flexible transformation such as a kernel density estimator would have been more insightful”**: We fully agree that the more flexible flow-based transformation outperforming the fixed transformations comes as no surprise and we did not intend to present this as a surprising result. Rather, our goal with this analysis was to provide explicit comparisons to existing models that would often use such fixed transformations, and to frame our approach based on normalizing flow as a natural extension of these methods. For example, fixed transformations are commonly used in the literature as variance-stabilizing transformation of neural responses which allows practitioners to use Gaussian-based models such as Factor Analysis on neural responses [1]. The direct comparison establishes that such fixed transformations can be improved by allowing the transformation to be learned directly from the data. Although one could derive the transformation function based on more flexible statistical techniques, such as kernel density estimator against responses to the repeated presentation of the stimulus, fitting a marginal density estimator for each stimulus would require repeats and it was our specific design goal to make the model trainable without the necessity for such repeats in the dataset.
>
>
> [1] Byron, M. Yu, et al. "Gaussian-process factor analysis for low-dimensional single-trial analysis of neural population activity." Advances in neural information processing systems. 2009.

---

### Official Review · Reviewer_BNcm · 2021-07-15

**Rating:** 8
**Confidence:** 5

**Summary:**

The authors present a Factor Analysis + flow model for modeling stimulus dependent neural responses. The FA + flow architecture allows the model to capture stimulus dependence, trial to trial correlated variability, and some features of the conditional and marginal response distributions. Validation is performed on synthetic data and real data from mouse visual cortex using 2 photon calcium imaging. The networks and inferred latent states are then compared to anatomical information and behavioral states.

**Ethical Concerns:**

The authors discuss some ethical concerns and contributions of improved modeling (including this method) to the use of animal models.

**Limitations And Societal Impact:**

The authors provide some discussion of the limitations. I think the limitations of diffeomorphisms could be more clear. Is another way of stating it that the diffeomorphisms cannot map a distribution with 1 peak (Gaussian) to a distribution with potentially multiple peaks (neural response)?

**Main Review:**

The paper is clearly written and the results and figures are clear. The FlowFA model is a concise way of including both stimulus dependent and stimulus conditioned (correlated) variability into a model. The exploration of potential uses of this type of methods in Fig 4 is also compelling (although see the technical concern). I also think this model bridges the gap between shallow (Factor Analysis) and deep (unconstrained flow models) in a way that provides both interpretability and more accurate modeling which could serve as useful techniques in future models.

I have one technical concern about how some the latent state results are interpreted, some other clarifying questions/comments about the method and results, and a suggestion for the discussion.

- I think it would benefit a broad neuroscience audience to spend a little more time describing flow-based models in Section 2.1. Contrasting them with standard feed forward models which are probably more familiar might help, i.e., what mappings can you do with flow based models (like inverse) that is not available with regular models. You mention exact likelihood computation and sampling, but expanding on this would help.
- Providing a bit more information about the neural data in Section 2.4 would help. How is the response variable being preprocessed? I would assume is it florescence $\Delta F / F$, but this isn't specified. Do you truncate it to $>=0$?
- In Fig 3b, what latent dimensionality was used? Do you get qualitatively similar transformations as a function of latent dimensionality?
- Fig 4a, in the Cortex Position panels, it would be helpful to explicitly state what color indicates in the Cortex positions and RF positions panels in the figure caption. I'm assuming experimentally measured area assignment.
- For 4a, does the inclusion of correlated variability in the model improve the inferred estimates of $\delta$ compared to a model fit with $C=0$, i.e., don't take correlations into account but share parameters in the same way? This analysis isn't strictly necessary to perform, but it would strengthen the result.

Technical concern (and my main reservation for a higher rating)
- For Fig 4 b and c, $C$ is only determined up to a rotation in the latent space. Both the likelihood and equation for $\mathbf v$ are unchanged by taking $C\rightarrow CR$ for some rotation $R$. $CRR^TC^T=CC^T$ for the likelihood and the distribution of $\mathbf v$ is unchanged since $\mathbf z$ is isotropic. I believe this complicates the interpretation of Fig 4 b and c since it is not clear how sign relationships will be preserved under arbitrary rotations. There are schemes for defining a particular rotation [1] or a different measure of $C$ which is invariant to rotations could be shown, such as the leverage scores (see [2] for a definition).
- Due to the above point, it might make more sense to report the $R^2$ from regressing pupil dilation onto all latent states since the particular latent state basis is not meaningful.

1. Kaiser, H. F. (1959). Computer program for varimax rotation in factor analysis. Educational and psychological measurement, 19(3), 413-420.
2. Mahoney, Michael W., and Petros Drineas. "CUR matrix decompositions for improved data analysis." Proceedings of the National Academy of Sciences 106.3 (2009): 697-702.

- For the results in Fig 4d, specify how $p(z|r)$ is computed in the FlowFA model (maybe earlier in section 2.1). Line 258 implies a MAP estimate is used, but it is not clear.
- For the sentence on lines 127-129, are only the nonlinearities shared but affine parameters different per-neuron? or all all nonlinearities and affine parameters shared? If the later, did you consider a model where each neuron has an independent set of parameters? Was overfitting a problem? Do you get roughly the same transformation for every neuron anyway?


This is somewhat tangential to the main thrust of the paper, but the sentence that starts on line 297 in the Discussion contrasts healthy, functional brains and brains with neurological and psychological disorders. People with neurological or psychological disorders can have healthy, functional brains. For example, rates of depression in graduate students in the US have greatly increased during the pandemic, but the cause probably won't be found in the brains of the grad students.
https://www.nature.com/articles/d41586-020-02439-6

I would rephrase this sentence to lessen the contrast. Suggestion:
"Accurate models of neural variability such as the one presented here can lead to deeper scientific insights and understanding of how brains perceive and compute with sensory information, and can eventually also provide insights into neurological and psychological disorders."

Minor comments
- and this cannot -> and thus cannot

**Time Spent Reviewing:**

5

---

> ### Author Response · Authors · 2021-08-10
> **Authors response to Reviewer BNcm**
>
> Thank you very much for your positive feedback. We are confident that we can address all of your concerns. Below is a detailed response to the points raised in your review:
>
>
> - **Re: Spend a little more time describing flow-based models**: We will improve the description of the flow-based models to be more readable and easily understandable.
>
>
> - **Re: Providing a bit more information about the neural data**: Briefly, the responses correspond to deconvolved Calcium traces using the constrained non-negative matrix factorization (CNMF) algorithm [1]. The neural responses for each stimulus image were then extracted as the accumulated inferred spiking rates of each neuron between 50 and 550 ms after the stimulus onset. We will provide additional description about the neural data in the revised manuscript.
>
>
> - **Re: Fig 3b, what latent dimensionality was used**: In this case the model with 3 latent dimensions was used, but we found that the learned transformations across models with different latent dimensions are nearly identical. We will add the relevant information in the caption of Figure 3, and add a figure in the supplementary to show the learned transformation across all models with different latent dimensions.
>
>
> - **Re: explicitly state what color indicates in the cortex positions and RF positions panels**: As you inferred correctly, the color in Fig 4a specifies the experimentally identified brain area. Importantly, the process of area identification was completely independent of the recording of the responses to natural images, which is the data we used for training our models. We will make sure that this information is clearly stated in the caption and text.
>
>
> - **Re: Does the inclusion of correlated variability in the model improve the inferred estimates of δ compared to a model with C=0?**: This would be an interesting analysis. However, it would require independently measured ground truth RF positions for all neurons, which, unfortunately, our dataset does not provide. Thus we cannot quantify the quality of estimated RF positions via the model. To assess whether the model has learned meaningful RF positions, we relied on the analysis of the mapping learned by our model to map Cortex positions to RF positions. As Figure 4a panel *det(Jacobian)* illustrates, our model does correctly pick up the known spatially mirrored retinotopy of different brain areas, which supports the accuracy of the inferred locations. To assess how the inclusion of correlated variability affects the quality of mapping, we quantified how well our model can classify distinct brain areas (different colors in Fig 4a panel *det(Jacobian)*) with different numbers of latent dimensions. The result shows that, on average across latent dimensions and random seeds, our model correctly classifies distinct brain areas with accuracy of 84% and 75% for scan 1 and scan 2, respectively, with no particular trend with respect to number of latent dimensions. More specifically, the classification accuracies for 0-, 1-, 2-, and 3-dimensional ZIFFA model were 85%, 88%, 79%, 85%, and 84% for scan 1, and for scan 2 the accuracies were 71%, 82%, 74%, 75%, and 71%. Please also note that the process of identifying brain areas is done by visual stimulation (dot mapping) and can be quite coarse. Thus, the measured “ground truth” areas might not be entirely correct around the area borders. It could thus be that our model provides a better estimate of the areas than the area assignment provided by dot mapping. Unfortunately, we have no way to directly verify this at the moment.
>
>
> - **Re: Rotational redundancy of $C$**: You are absolutely right that $C$ is only determined up to a rotation, which seems to make the interpretation of Fig 4b and 4c arbitrary. To address this issue, we will follow a similar approach to Yu et. al 2009 [2] which is akin to the varimax rotation in the first reference you pointed out: we can look at the problem from the view that the columns of $C$ can have arbitrary direction and scale, therefore the inferred latent states can be arbitrarily interpreted. One solution to this is to orthonormalize the columns of $C$. That is, change columns of $C$ such that instead of representing an arbitrary axis in the high-dimensional neural space, they represent a set of basis vectors (i.e. set of unit-length vectors that are orthogonal to each other). We achieve this orthonormalization by applying the singular value decomposition to the learned $C$ which yields $C = UDV'$. As a result $Cz$ can be re-written as $Cz = U(DV'z) = U\tilde{z}$ where $\tilde{z}$ is the *orthonormalized latent state*. Consequently, instead of visualizing the MAP of $z$, $\mathbb{E}[\mathbf{z} | \mathbf{x}, \mathbf{r}]$, we would visualize $DV' \mathbb{E}[\mathbf{z} | \mathbf{x}, \mathbf{r}]$. An important advantage of this approach is that, while the elements of $z$ (and corresponding columns of $C$) have no particular order, the elements of $\tilde{z}$ (and corresponding elements of $U$) are ordered by the amount of data covariance explained. Therefore, the inferred latent states are ordered by their contribution in explaining the covariance observed in neural activity, resulting in more intuitive and interpretable latent states. We will perform this extended step and change Fig 4 such that it uses the *orthonormalized latent state* $\tilde{z}$ and the corresponding factor loading matrix $U$. Performing this analysis on our latest model (ZIFFA; see general response) revealed an interesting observation: among the *orthonormalized latent states* $\tilde{z}$, one dimension, compared to other dimensions, has a significantly higher $R^2$ value (scan 1: 0.53; scan 2: 0.63) with pupil dilation. Interestingly, this latent dimension is the second dimension for both scans, while the first dimension explaining most of the shared variance among neurons has an $R^2$ value of 0 with pupil dilation, for both scans. We found this observation to be consistent across models initialized and trained with different random seeds.
>
>
> - **Re: $R^2$ from regressing pupil dilation onto all latent states**: We think that our solution for the rotational redundancy of $C$ also addresses this point (please see the previous response regarding the rotational redundancy of $C$). However, for completeness sake, we also regressed the pupil dilation onto the latent states. After performing the regression analysis on the latent states inferred from our latest model (ZIFFA) with 3 latent state dimensions, we observe that the $R^2$ value between the pupil dilation and the regressed output is 0.56 (P < 0.001, two-tailed test for significance of correlation) and 0.76 (P < 0.001) for scan 1 and scan 2, respectively. The $R^2$ values between pupil dilation and each inferred latent state $z$ are 0.33, 0.25, and 0.08 for scan 1; and 0.09, 0.15, and 0.55 for scan 2.
>
>
> - **Re: How is p(z|r) computed in the FlowFA model**: $p(\mathbf{z} | \mathbf{r})$ is indeed a MAP estimate. We will make sure this is stated clearly in the text (of section 2.1).
>
>
> - **Re: Shared vs per-neuron parameters of the affine transformation**: In the submitted version of the manuscript, all nonlinearities and affine parameters were shared. However, we have since extended the model to allow for neuron-specific parameters and we show that this improves the model. In the final version of the manuscript we will add:
>     - A supplementary figure showing performance comparison for models with shared vs per-neuron Flow parameters. Briefly, learning a neuron-specific transformation generally results in an improved likelihood. For the FlowFA model, the average improvement, across all latent dimensions, in log-likelihood (measured in bits per image and per neuron) is 0.071 for scan 1 and 0.004 for scan 2. Similarly, learning neuron-specific transformations results in an improvement in log-likelihood for the ZIFFA model: 0.009 for scan 1 and 0.004 for scan 2.
>     - Updated version of Fig 2b to show the learned transformation for all neurons. The transformations for most of the neurons have roughly similar shapes but they are not exactly the same.
>
>
> - We appreciate your suggestion for improving line 297. We will rephrase the sentence based on your suggestion to avoid contrasting the functional brain and brain with psychological disorder.
>
>
> - Thank you for pointing out the typo. We will thoroughly check the paper for typos and naming inconsistencies.
>
>
> - **Re: discussion on the limitations of diffeomorphisms**: Sorry, that this point was unclear. In general, a diffeomorphism can map a unimodal into a multimodal distribution. For instance the mapping $CDF^{-1}_\text{MoG}(CDF_\text{Gauss}(\cdot))$ would map a Gaussian variable into a mixture-of-Gaussians-distributed variable (MoG) with multiple peaks. However, diffeomorphisms cannot map a delta peak into something that’s not a delta peak, since it cannot map multiple distinct values onto a single value due to its invertibility. Note that our modified model (ZIFFA) now avoids this problem by only transforming the positive part of the response with a diffeomorphism (flow).
>
>
> [1] Pnevmatikakis, Eftychios A., et al. "Simultaneous denoising, deconvolution, and demixing of calcium imaging data." Neuron 89.2 (2016): 285-299.
>
> [2] Byron, M. Yu, et al. "Gaussian-process factor analysis for low-dimensional single-trial analysis of neural population activity." Advances in neural information processing systems. 2009.

---

> > ### Comment · Reviewer_BNcm · 2021-08-17
> > **Response to authors**
> >
> > Thanks for addressing my questions and comments. Combines with the changes from the other reviews, I would change my score from a 7 to an 8.

---

> > > ### Author Response · Authors · 2021-08-17
> > > **Thanks!**
> > >
> > > Thanks a lot for the feedback! We are very happy to read this and to hear that we were able to clarify your questions.

---

> > > ### Author Response · Authors · 2021-08-30
> > > **Clarification on the change of rating**
> > >
> > > Dear Reviewer BNcm,
> > >
> > > Thanks again for your helpful comments and for your willingness to increase the rating from a 7 to an 8. While we are very happy to hear that you are willing to increase your rating, we realized that the rating under the main response has not been changed so far (i.e. the rating remained a 7). We were wondering whether you may still have open questions or whether you might just have forgotten to change the rating? If there are still open questions, please let us know. Thanks a lot.

---

### Official Review · Reviewer_QQF7 · 2021-07-17

**Rating:** 7
**Confidence:** 4

**Summary:**

This paper proposes a model of neural responses by capturing both stimulus dependent mean as well as variance. They transform the firing rate of each neuron using a reversible flow based model and model the transformed responses as a multi-variate gaussian. The mean of the multivariate gaussian is given by a neural network transformation of the input stimulus, and the low rank + diagonal covariance is independent of stimulus. They show the utility of this approach on synthetic as well as calcium imaging recordings from mouse visual cortex.

**Ethical Concerns:**

no ethical cconcern.

**Limitations And Societal Impact:**

no societal impact.

**Main Review:**

The paper is well written, performs relevant baseline comparisons and shows the utility on real neuroscience data.

* I think the overall idea is quite novel. It might be helpful to describe the approach as learning one transformation of stimulus and another transformation of responses, and mapping them both to a common space where geometry is simpler. This general idea is quite different than most models in computational neuroscience that either learn encoding (stimulus-> response), or decoding (response-> stimulus) models.

* My high-level comment is that the ideas can be simplified by removing the connection to flow models at many places. The paper uses (1) only per-neuron invertible transformations and (2) uses same transformation for each neuron, thereby not using the full power of current flow-based models.

* In terms of an ideal model, I think the main limitation is that the correlated noise in responses does not have a clear dependence on stimulus. That is, the degree of covariance between responses could be high for some stimuli and low for others in neuroscience data, but the model does not capture that. This limitation should be highlighted better in the text.

* It seems that the transformation learned by the flow in Figure 3b is highly nonlinear and step-like (pink line). Would it be totally fine to lose the intertibility property of flow models then? All components of the model are not necessary for this real data case.

* Similarly, the ZIG model outperforms FAFLOW model for zero latents. Losing the invertibility of FLOW model for a transformation that looks more like ZIG (Figure 5) might be better overall (even for >0 factors).

* The conditional correlation plots in Figure 3a are not explained in the main text, and they seem to be the strongest results.
* Moreover, why does flow model perform better than ZIG for 0 latents (where cells are modeled independently) for conditional correlation metric, but not for likelihood metric?

* The interpretation of latents (Figure 4) could be cofounded by experimental factors. The spatial smoothness of latents could be explained by spatially smooth expression levels of calcium indicator, or spatially smooth recording noise. Either control for these confounds, or explain them as possible interpretation.

* The det(Jacobian) explanation in Figure 4 and text is not clear. Please explain them in non-mathematical terms to highlight the relevance/importance.

* Color bars missing in Figure 4

**Time Spent Reviewing:**

6 hours

---

> ### Author Response · Authors · 2021-08-10
> **Authors response to Reviewer QQF7**
>
> Thank you very much for your helpful feedback. We performed additional analysis and comparisons of the models, and provided further explanation to make some of the claims and descriptions in the manuscript more clear.
>
> - **Re: Describing the approach as learning one transformation of stimulus and another transformation of responses**: We appreciate your suggestion. Indeed one of the goals for Figure 1 was to introduce the overall approach in such light. We will make sure to clearly and intuitively explain the contrast between our approach and common models in computational neuroscience.
>
>
> - **Re: Removing the connection to flow models**: Our goal is to develop an accurate probabilistic model of neural responses to natural stimuli $p(\text{responses} \mid \text{stimulus})$. To achieve this in our model, a crucial aspect is the invertibility of the transformation used in the model, which allows us to 1) easily apply known latent variable models (i.e. Factor Analysis) on the neural responses, 2) evaluate the full likelihood of the responses, and 3) sample and simulate neural responses to novel stimuli. This makes our model one instance of a larger family of models that use flow-based transformations on neural responses. Without the invertibility, the direct evaluation of the likelihood, and thus the ability to fit the model with maximum likelihood, would become infeasible in most cases. To address your concern, we will simplify the explanation by clearly explaining 1) the nature of the flow-based transformation used in this work, 2) contrast it with common full-fledged Normalizing Flow models, and 3) discussing the advantages and disadvantages of using marginal vs. full-fledged Flow models (such as the ability to compute conditionals, which we already discussed in section 1, line 74, in the original manuscript).
>
>
> - **Re: Correlated noise in responses does not have a clear dependence on stimulus**: As the reviewer pointed out, the learned covariance structure on the “transformed” neural responses captured by the Factor Analysis (FA) model indeed does not vary with the stimulus, and the stimulus is only used to shift the mean of the FA model. However, this is not true for the original “untransformed” neural responses since the nonlinear flow transformation actually can introduce changes in the covariance as the mean varies. We demonstrate this effect in Fig 2a. Specifically, samples drawn from 2-dimensional Gaussian distributions with differing means (indicated by the color of the samples) but identical covariance exhibit different covariances after being transformed via the nonlinear transformation functions. This mean-dependent changes in the covariance potentially allows us to capture changes to the noise correlation based on stimulus.  That being said, a possible extension of our model is an explicit dependence of the FA’s covariance on the stimulus. We will include more through discussion about this point in the revised manuscript.
>
>
> - **Re: a) the step-like shape of the flow-based transformation and b) whether we can “lose the invertibility property”**:
>     - a) The step-like shape reflects the bias towards the “peak” at zero in the original response distribution. A significant portion of “de-convolved” responses recorded via two-photon Calcium imaging are zero, resulting in a zero-inflated response distribution. The learned transformation by the FlowFA model attempts to generate this peak by mapping a large proportion of the Gaussian probability mass onto the  “zero” responses. This is clearly an undesirable property. As we pointed out in our general response above, we have fixed this issue by now applying the flow transformation to the non-zero responses only. We call this extended model Zero-Inflated Flow-based Factor Analysis (ZIFFA). Importantly, while ZIFFA preserves all properties of the FlowFA model, it a) captures the marginal distributions more accurately, b) achieves a higher likelihood, and c) learns less step-like transformations. We will add the results from the ZIFFA model in the final version of our manuscript.
>     - b) As mentioned in the response **Re: Removing the connection to flow models**, the invertibility of the transformation is a necessary property in our model to - among other properties - obtain a tractable likelihood that can be optimized.
>
>
> - **Re: ZIG model outperforms the FlowFA model for zero latents**: The ZIG indeed outperforms FlowFA for zero latent. However, the ZIFFA (see general response above) model outperforms the ZIG model in all latent dimensions including zero by capturing the marginal distributions more accurately. More specifically, for 0 latent dimensions the log-likelihood of ZIFFA vs ZIG is 8.87 vs 8.84 (measured in bits per image per neuron; mean across random seeds, P < 0.01, Wilcoxon rank-sum test) for scan 1, and for scan 2 the log-likelihoods are 11.31 vs 11.30 (mean across random seeds, P < 0.01, Wilcoxon rank-sum test). Moreover, with increasing latent dimensions, the performance of the ZIFFA model improves: 1-, 2-, and 3-dimensional latent state results in log-likelihoods of 8.88, 8.89, and 8.90 for scan 1, and 11.322, 11.329, and 11.334 for scan 2.
>
>
> - **Re: The conditional correlation plots in Figure 3a are not explained in the main text**: The computation of conditional correlation is explained in section 2.6, and the results are briefly mentioned in 3.1. We will update section 3.1 and mention the results of Fig 3a more prominently.
>
>
> - **Re: Why does the flow model perform better than ZIG for 0 latents for conditional correlation metric, but not for likelihood metric?**: When the number of latent dimensions is 0, both ZIG and FlowFA models assume independence between neurons (FlowFA would have a diagonal covariance matrix). As a result, the FlowFA model cannot take advantage of conditioning (as opposed to latent dimensions >0) and the models are comparable. In this case (where we have 0 latent dimensions), the FlowFA model has a rather similar correlation as ZIG. However, the ZIG model seems to capture the marginal distribution of responses better than FlowFA, therefore yielding a higher likelihood compared to FlowFA model. Our extended ZIFFA model (see general response), now captures the marginal distributions more accurately than both FlowFA and ZIG models (refer to the table in general response for the performance comparison).
>
>
> - **Re: The interpretation of latents (Figure 4) could be cofounded by experimental factors**: Here, we simply wanted to point out the utility of the presented model in facilitating biological insights, but did not intend to imply any conclusive biological interpretations in this figure. You are absolutely right that a biological conclusion would need more careful investigation that takes alternative (possibly confounding) explanations into account. As the main focus of the paper is the presentation of the model, this would go beyond the scope (and space limit) of the paper. We will make sure that other possible interpretations of the results shown in Figure 4 are clearly stated in the text.
>
>
> - **Re: Explanation of the det(Jacobian)**: Several visual areas in mice show retinotopies that are “flipped” with respect to each other [1]. Intuitively, this means that if we imagine a point moving along the cortical surface, its counterpart in visual space would reverse its movement direction as we cross the boundary between two areas with flipped retinotopies with respect to each other. An equivalent way to see this is that, as we cross the boundary, one tangent vector of the mapping from cortex to visual space flips. As a result, the determinant will flip its sign. This will also happen if the x and y direction on the cortex are not perfectly aligned with x and y in visual space. This is captured by the determinant of the Jacobian. In order to communicate this more clearly, we will use *model-based area identification* as a replacement of *det(Jacobian)* in the title of the corresponding panel in Figure 4a. Furthermore, we will add a more detailed explanation about the intuition behind the method and its relevance/importance in the final version of the manuscript.
>
>
> - **Re: Missing color bars in Figure 4**: thank you for pointing this out. We will add the color bars as well as other clarifying information to the figure in the revised version of the manuscript.
>
> [1] Garrett, Marina E., et al. "Topography and areal organization of mouse visual cortex." Journal of Neuroscience 34.37 (2014): 12587-12600.

---

> > ### Comment · Reviewer_QQF7 · 2021-08-23
> > **Thanks for detailed responses**
> >
> > Thank you for the detailed responses. With the modified model that performs better, I would increase my score from 6 to 7.

---

> > > ### Author Response · Authors · 2021-08-23
> > > **Thank you!**
> > >
> > > Thank you very much for the feedback! We are very glad to hear that we were able to clarify your concerns.

---

### Author Response · Authors · 2021-08-10
**Authors general response**

We would like to thank the reviewers for their very helpful comments and feedback on our manuscript. We were happy to see that the reviewers found our manuscript to be “well written” (Rev **QQF7**) and the results to be “clear” (Rev **BNcm**) and “novel” (Rev **QQF7**, **mWE7**), “bring new Machine Learning concepts […] into computational neuroscience” (Rev **Two8**) where “models for taking both stimulus driven and stimulus-conditioned activity into account are important tools” (Rev **mWE7**), remarking that our model “is a concise way of including both stimulus dependent and stimulus conditioned [...] variability into a model” that “bridges the gap between shallow (Factor Analysis) and deep (unconstrained flow models) in a way that provides both interpretability and more accurate modeling which could serve as useful techniques in future models” (Rev **BNcm**).

The main points raised by reviewers were:
1. The recommendation to provide a more detailed explanation of the flow-based models, specifically, to contrast it with the standard feed-forward models found in system-identification (Reviewers **QQF7** and **BNcm**).
2. Questions about details regarding the neural responses, asking for more information about the neural data (Reviewers **BNcm**, **mWE7**, and **Two8**).
3. The request for an expanded discussion on the limitations of the model (Reviewers **BNcm** and **mWE7**).
4. Concerns over the fact that our flow-based model (FlowFA) was outperformed by the ZIG model that assumes independence (Reviewers **QQF7** and **mWE7**).
5. Questions to what extent our FA covariance could capture the stimulus-dependent noise correlations (Reviewers **QQF7** and **Two7**).
6. Limited comparisons to existing models (Reviewer **mWE7**).
7. Questions about the analysis of the trained model and the possible biological implications of the inferred latent states (Reviewers **QQF7**, **BNcm**, and **Two8**).

We are confident that we can address all of the above concerns. While we provide a summary here, please also refer to our more detailed responses to each individual review below.

**Regarding 1 - 3**: We have posted a detailed response to each individual reviewer, and will revise the manuscript accordingly.

**Regarding 4**: Since the initial submission, we have extended the flow transformation to allow the model to learn neuron-specific transformations, by learning neuron-specific parameters. Moreover, we have introduced a small change into our model to improve its performance and its ability to accurately capture the marginals (which we discussed on line 286 in the initial manuscript). Instead of transforming the entire neural response with the flow transform, we only transform *positive* responses. In essence, the zero part is now modelled with a separate peak (similar to ZIG) while the *joint* positive response is modelled with a FlowFA model. While this small change leaves all properties of the model intact, it improves the performance (now outperforming ZIG) and yields much better fits to the marginals. To make this change apparent, we refer to the new model as Zero-Inflated Flow-based Factor Analysis (ZIFFA) and will update the manuscript accordingly. For reference and to demonstrate that this change addresses point 4, we provide a table comparing our FlowFA model, ZIG, and ZIFFA.

| Model       | Log-likelihood (scan 1) | Log-likelihood (scan 2) |
| ------------- |:-------------:|:-------------:|
| ZIG     | 8.84 | 11.30 |
| FlowFA (0-d latent state) | 8.48 | 10.95 |
| FlowFA (1-d latent state) | 8.55 | 11.01 |
| FlowFA (2-d latent state) | 8.59 | 11.04 |
| FlowFA (3-d latent state) | 8.63 | 11.07 |
| ZIFFA (0-d latent state) | 8.87 | 11.310 |
| ZIFFA (1-d latent state) | 8.88 | 11.322 |
| ZIFFA (2-d latent state) | 8.89 | 11.329 |
| **ZIFFA (3-d latent state)** | **8.90** | **11.334** |


**Regarding 5**: We addressed this point by 1) explaining how the current model does result in a dependency between the mean and variance of neural responses due to the flow transformation, 2) explaining how the model can be extended to directly model stimulus-dependent noise correlation, and 3) performing additional analysis assessing how much the empirical noise correlation depends on the stimulus. The result of this analysis shows that noise correlations in our dataset do not exhibit observable dependence on the stimuli, which in turn suggests that there is no meaningful dependence of the noise correlations on the stimuli to be captured by our model. In the final version of the manuscript we will discuss the limitations of the model more clearly and add this analysis and the corresponding results.

**Regarding 6**: Our goal is to have a stimulus-dependent likelihood model given the stimulus. While there are many models that can capture stimulus-conditioned variability in the neural responses, many of them rely on repeated presentations of the same stimulus to inform the model. In contrast, we provide a model that can be trained end-to-end with a feasible likelihood on single presentations of natural images only and be tested to predict response distributions on entirely novel stimuli. For this reason, a direct performance-based comparison is unfortunately difficult without substantial extensions of previous approaches, which goes beyond the scope of the paper. For the model where such a comparison is feasible (ZIG) we did already provide a quantitative comparison. For the other suggested models, we will include an extended discussion about the commonalities and differences to our flow-based approach (please see responses to individual reviewers).

**Regarding 7**: We addressed the concerns about the analysis of the trained model and the inferred latent states. More specifically,
- We assessed the quality of the learned mapping between cortex positions and receptive field (RF) positions by quantifying how well our model can classify distinct brain areas. The results show that our model can classify brain areas with accuracy of 84% and 75% for scan 1 and scan 2, respectively. This is likely an underestimation, as the ground truth area assignment might not be 100% accurate due to experimental precision.
- We performed regression analysis on the inferred latent states and pupil dilation, suggested by Reviewer **BNcm**, which demonstrates that the $R^2$ value between latent state and pupil dilation is statistically significant (P < 0.001) and it lies well within the range reported previously in the literature.
- We added a more detailed explanation of the concepts that were found confusing (e.g. det(Jacobian)) and will revise the manuscript accordingly.

We provided more details in our responses to the individual reviewers, outlining the analysis performed to address the concerns and the steps we would take to improve on the final revision of the manuscript. This is a list of the analyses we performed to address the points raised by the reviewers:
1. We demonstrated that the increase in log-likelihood with an increasing number of latent dimensions is statistically significant.
2. We computed empirical noise correlation of our dataset to assess and show that the empirical noise correlations do not exhibit observable dependency on the stimulus.
3. We quantified the quality of the mapping between cortex positions and RF positions.
4. We performed a regression analysis between inferred latent states and pupil dilation, and showed that the correlation between latent state and pupil dilation is statistically significant.
5. We extended the analysis of the latent state model to disambiguate the rotational redundancy of factor loading matrix $C$.
6. We assessed the learned flow transformation across different latent dimensions.
7. We quantified the performance of flow-based models with shared vs. per-neuron parameters of the flow transformation.

We hope that our responses address all potential concerns. Please let us know if you need any further clarifications.

---

### Decision · Program_Chairs · 2021-09-27

**Decision:**

Accept (Spotlight)

**Comment:**

This paper introduces a system identification model for stimulus-driven and stimulus-conditioned fluctuations. They propose an architecture based on factor analysis and a flow model, and validate their performance on synthetic and mouse visual calcium imaging data. The paper addresses an important question in neuroscience, it is well written, introduces a new ML approach to computational neuroscience, and the authors have done a great deal to address all the concerns of the reviewers and convince them of the points they intended to make. Given this, I recommend this paper for acceptance and ask the authors to make all the modifications they promised in the final manuscript.